# SND1 binds to ERG and promotes tumor growth in genetic mouse models of prostate cancer

Sheng-You Liao [1], Dmytro Rudoy[1], Sander B. Frank [1], Luan T. Phan[1], Olga Klezovitch[1], Julian Kwan[2], Ilsa Coleman[1], Michael C. Haffner[1,3], Dapei Li[4], Peter S. Nelson [1,3,4,5], Andrew Emili [2,6] & Valeri Vasioukhin [1,3] ✉

SND1 and MTDH are known to promote cancer and therapy resistance, but their mechanisms and interactions with other oncogenes remain unclear. Here, we show that oncoprotein ERG interacts with SND1/MTDH complex through SND1's Tudor domain. *ERG*, an ETS-domain transcription factor, is overexpressed in many prostate cancers. Knocking down *SND1* in human prostate epithelial cells, especially those overexpressing *ERG*, negatively impacts cell proliferation. Transcriptional analysis shows substantial overlap in genes regulated by *ERG* and *SND1*. Mechanistically, we show that ERG promotes nuclear localization of SND1/MTDH. Forced nuclear localization of SND1 prominently increases its growth promoting function irrespective of *ERG* expression. In mice, prostate-specific *Snd1* deletion reduces cancer growth and tumor burden in a prostate cancer model (*PB-Cre/Pten*^*flox/flox*/*ERG* mice), Moreover, we find a significant overlap between prostate transcriptional signatures of ERG and SND1. These findings highlight SND1's crucial role in prostate tumorigenesis, suggesting SND1 as a potential therapeutic target in prostate cancer.

Comprehensive genome-wide characterization of human malignancies identified many genes that are recurrently altered in cancer. The exact function of these genes in normal cells and in cancer are often poorly understood. However, such knowledge will be necessary in the future for the development of therapeutic interventions that target tumors harboring specific genomic modifications. The most common genetic alterations in human prostate cancer (PC) are gene fusions involving the androgen-regulated *TMPRSS2* gene and the coding sequences of a member of the ETS family transcription factor *ERG*, which occur in approximately half of all human prostate cancers[1–3]. The *TMPRSS2-ERG* fusion is not only a cancer initiating event, but also required for the survival of ERG-expressing PC cells[1–3]. High expression of *TMPRSS2-*

*ERG* fusion is maintained in advanced stage and metastatic tumors[4]. The mechanism responsible for ERG-mediated prostate cancer are not well understood. ERG has been implicated in the regulation of Wnt, NF-κB, TGF-beta, EZH2, Notch, ERF, ETS2, Hippo, BAF chromatin remodeling complexes, and androgen receptor (AR) activity[5–17]. Even though ERG was found to regulate multiple cancer relevant signaling pathways, the exact mechanism responsible for ERG-meditated oncogenic transformation at the time of human PC initiation is still not clear.

SND1 and its binding partner MTDH are two proteins that are prominently implicated in cellular transformation, cancer metastasis, and drug resistance[18–21]. SND1/MTDH are frequently overexpressed and

[1]Division of Human Biology, Fred Hutchinson Cancer Center, Seattle, WA, USA. [2]Center for Network Systems Biology, Departments of Biochemistry & Biology, Boston University, Boston, MA, USA. [3]Department of Laboratory Medicine and Pathology, University of Washington, Seattle, WA, USA. [4]Department of Medicine, Division of Medical Oncology, University of Washington, Seattle, WA, USA. [5]Division of Clinical Research, Fred Hutchinson Cancer Center, Seattle, WA, USA. [6]Division of Oncological Sciences, Knight Cancer Institute, Oregon Health & Science University, Portland, OR, USA. ✉e-mail: vvasiouk@fredhutch.org

associated with poor prognosis across many cancer types including PC[22–26]. MTDH can activate several signaling pathways, such as PI3K/AKT, NF-κB, Wnt, and MAPK which are involved in cell transformation, proliferation, invasion, and angiogenesis[23,27–29]. The interaction between MTDH and SND1 is critical for its oncogenic function in breast cancer and disruption of this interaction results in suppression of cancer progression and metastasis[30–33]. SND1, also called Tudor-SN or p100, is a multifunctional protein containing multiple nuclease and modified Tudor domains[26]. SND1 is a component of the RNA-induced splicing complex (RISC) that mediates RNA interference[34]. SND1 cooperates with MTDH to form the RISC to facilitate degradation of tumor suppressor mRNAs thus promoting liver cancer cell proliferation[35]. SND1 mediates mature miRNA decay[36]. In addition, SND1 also functions as transcriptional co-activator[37,38]. While the important role of SND1/MTDH in cancer is well documented, the molecular mechanisms of their function are only beginning to be understood.

In this study, we find that ERG interacts with SND1 and the entire SND1/MTDH protein complex in prostate epithelial cells and we identify the ERG and SND1 domains responsible for this interaction. SND1 is necessary for ERG-mediated transformation of human prostate epithelial cells. To analyze the significance of this interaction in vivo we generated mice with a conditional allele of *Snd1*. We find that *Snd1* is necessary for the growth of PC in mice with prostate-gland specific overexpression of *ERG* and inactivation of *Pten*. Mechanistically, we find that ERG upregulation increases nuclear localization of SND1/MTDH and nuclear localized SND1 promotes PC cell transformation independently of ERG overexpression. Overall, these findings reveal an additional mechanism of ERG function in PC and identify SND1 as an important factor that contributes to PC initiation and progression.

## Results

### ERG interacts with the SND1/MTDH protein complex

To obtain insights into the molecular mechanisms responsible for ERG-mediated transformation of prostate epithelial cells, we performed an Affinity Purification–Mass Spectrometry (AP/MS) study of epitope-tagged ERG. For this purpose, we expressed N-terminal and C-terminal tagged-ERG in the human PC cell line VCaP, which harbors an endogenous *TMPRSS2-ERG* gene fusion event[1]. VCaP cells expressing tagged-GFP were used as a negative control. High precision mass spectrometry analysis identified 216 putative ERG interacting proteins, which were found in pull-downs with both N-terminal and C-terminal epitope-tagged-ERG, but not with tagged-GFP control (more than one unique peptide for each) (Fig. 1a, b, Supplementary Data 1). While our unbiased AP/MS experiments identified multiple not previously reported ERG interacting proteins, we were most intrigued by the binding between ERG and SND1/MTDH protein complex and focused on characterizing the potential oncogenic roles for these interactions in more detail (Fig. 1c).

Protein interactions between ERG and SND1/MTDH were first confirmed by co-expression and co-immunoprecipitation experiments in HEK293 cells (Fig. 1d). Next, we validated these interactions by co-immunoprecipitation using endogenously expressed ERG and SND1/MTDH proteins in VCaP cells (Fig. 1e). In addition, to confirm the in situ interaction and determine the subcellular localization of ERG-SND1/MTDH binding in VCaP cells we performed in situ proximity ligation assays (PLAs). We found that ERG interacted with SND1 in both the nucleus and cytoplasm (Fig. 1f). Interestingly, immunofluorescence staining for total ERG and SND1/MTDH proteins revealed that the majority of SND1/MTDH were present in the cytoplasm and only a small fraction of SND1 is present in the nucleus in these cells (Supplementary Fig. 1). In contrast, the majority of ERG was present in the nucleus and only a small proportion was present in the cytoplasm. Overall, we concluded that endogenous ERG and SND1/MTDH proteins stably and specifically interact with each other in both the cytoplasm and nucleus of human PC cells.

### The Tudor domain of SND1 and the N-terminal domain of ERG are involved in interactions between ERG and SND1/MTDH protein complexes

SND1 and MTDH proteins are tightly bound to each other[30]. To determine whether SND1 or MTDH is primarily responsible for the interaction between ERG and SND1/MTDH, we performed siRNA for endogenous SND1 or MTDH followed by co-immunoprecipitation experiments. Knockdown of endogenous SND1 in VCaP cells with two independent siRNA oligos resulted in decreased amounts of ERG pulled down by anti-MTDH antibodies (Fig. 2a). In contrast, knockdown of endogenous MTDH did not decrease the amounts of ERG pulled down by anti-SND1 antibodies (Fig. 2b). Moreover, in co-expression/co-immunoprecipitation experiments using overexpressed tagged proteins in HEK293 cells, HA-ERG pulled down MTDH only in the presence of co-expressed SND1, while SND1 was efficiently pulled down by ERG without overexpression of MTDH (Fig. 2c). These data together indicate that SND1 is the primary binding partner of ERG in the ERG-SND1/MTDH interaction.

To identify the domain(s) of ERG and SND1 involved in ERG-SND1 binding, we performed co-immunoprecipitation experiments using a panel of recombinant ERG and SND1 fragments expressed in HEK293T cells. The N-terminal region of ERG showed strong binding to SND1 (Fig. 2d). Expression of individual ERG domains revealed that the N-terminal domain and to a lesser degree the ETS domain in the C-terminus of ERG interacted with SND1 (Fig. 2e, f). In turn, co-immunoprecipitation experiments with a panel of recombinant SND1 proteins revealed that the Tudor Domain of SND1 is primarily responsible for interaction between SND1 and ERG (Fig. 2g–i).

### SND1 is necessary for ERG-mediated promotion of cell proliferation in human prostate epithelial cells

Expression of exogenous ERG in immortalized non-tumorigenic human prostate epithelial RWPE-1 cells increased the size of 3D colonies growing in organoid culture conditions in the presence of matrigel (Fig. 3a–c). To determine whether *SND1* or *MTDH* are necessary for ERG-mediated increases in organoid size, we knocked down endogenous *SND1* or *MTDH* using two independent shRNA constructs for each gene (Fig. 3a, Supplementary Fig. 2a). While the knockdowns of *SND1* or *MTDH* did not impact the size of the colonies in control RWPE-1-GFP cells, they significantly decreased the size of colonies in RWPE-1-ERG cells (Fig. 3b, c, Supplementary Fig. 2b).

To further corroborate these findings we used a CRISPR/Cas9 system to generate RWPE-1-GFP and RWPE-1-ERG cell lines with the knockout of both alleles of endogenous *SND1* and then generated cells expressing either exogenous V5-SND1 or V5-GFP control (Fig. 3d). We found that in both 3D and 2D systems the ERG-mediated increase in the colony size was observed in cells re-expressing SND1 (Fig. 3e, f, Supplementary Fig. 3a). We concluded that SND1 is necessary for ERG-mediated increase in colony size in RWPE-1 cells.

Endogenous ERG is prominently overexpressed in the human PC cell line VCaP[1]. As expected, siRNA-mediated knockdown of ERG in these cells resulted in decreased matrigel cell invasion and proliferation (Fig. 3g–i). We found that the knockdown of endogenous *SND1* with two independent siRNAs also decreased VCaP cell invasion and proliferation (Fig. 3g–i). Similar results were obtained using the ERG-expressing human PC cell line LuCaP 35CR (Supplementary Fig. 3b, c). Interestingly, in both models the combined knockdown of *ERG* and *SND1* did not show an additive effect suggesting that ERG and SND1 are likely functioning in the same pathway that regulates the proliferation of ERG-positive human PC cells and elimination of either of them is sufficient to disrupt this signaling pathway (Fig. 3j and Supplementary Fig. 3b, c).

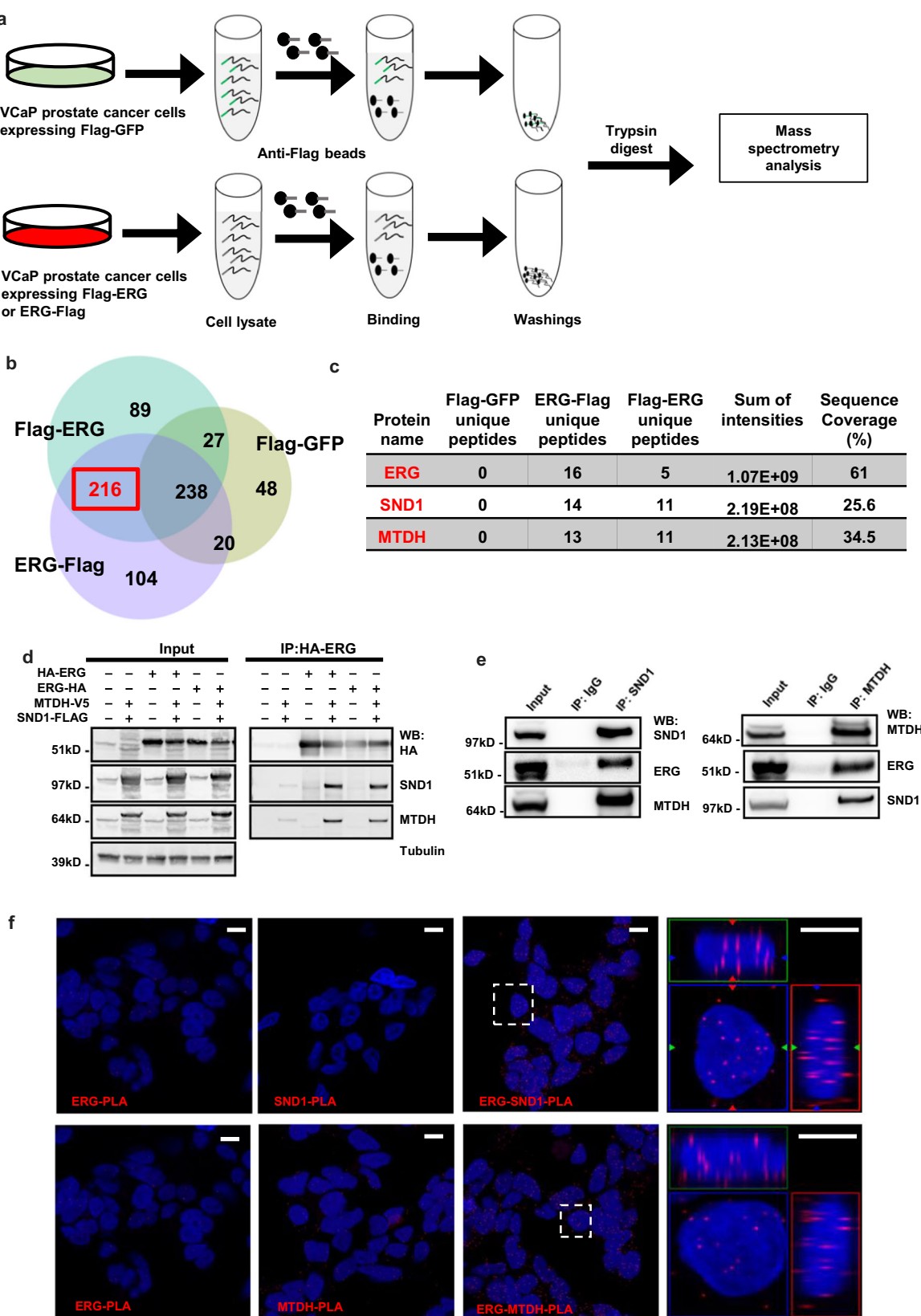

In addition to loss-of-function, we also performed gain-of-function experiments. Overexpression of *SND1* increased the proliferation of VCaP cells (Fig. 3k, l). Overexpression of *SND1* in VCaP cells with knockdown of endogenous ERG resulted in attenuated pro-growth phenotype, again indicating that both ERG and SND1 promote proliferation of VCaP cells (Fig. 3l).

## *SND1* and *ERG* regulate similar but not identical transcriptional programs in prostate epithelial cells

ERG is known to function primarily as a transcription factor and SND1 (also known as transcriptional co-activator p100) has also been implicated in regulating the transcription and the stability of various mRNAs[36,37,39]. To analyze the role of both *ERG* and *SND1* in VCaP cell

**Fig. 1 | ERG interacts with the MTDH/SND1 protein complex in PC cells.**
**a** Schematic of IP-mass spectrometry experiment in VCaP PC cells expressing epi-tope-tagged-GFP, N-terminal or C-terminal epitope-tagged-ERG. **b** Venn diagram showing the overlaps of IP-mass spectrometry identified proteins. **c** IP-mass spec-trometry data from experiments in (**a**, **b**) for ERG, SND1, and MTDH proteins. **d** Co-immunoprecipitation (IP) of epitope-tagged ERG, MTDH, and SND1 proteins expressed in HEK293 cells. **e** Co-IP of endogenous ERG, SND1, and MTDH proteins

from VCaP human prostate epithelial cells. **f** Confocal images of proximity ligation assay (PLA) of endogenous ERG, SND1, and MTDH proteins in VCaP cells. An interaction or close proximity between two proteins is revealed by the appearance of red fluorescent spots. DAPI (blue) indicates nuclear stain. Areas indicated by dashed white squares are shown at higher magnification as 3-dimentional projec-tions. kD, kilodalton. Scale bar, 10 μm. Experiments were repeated 2 (**d**), 3 (**e**), or 6 (**f**) times with similar results. Source data are provided as a Source data file.

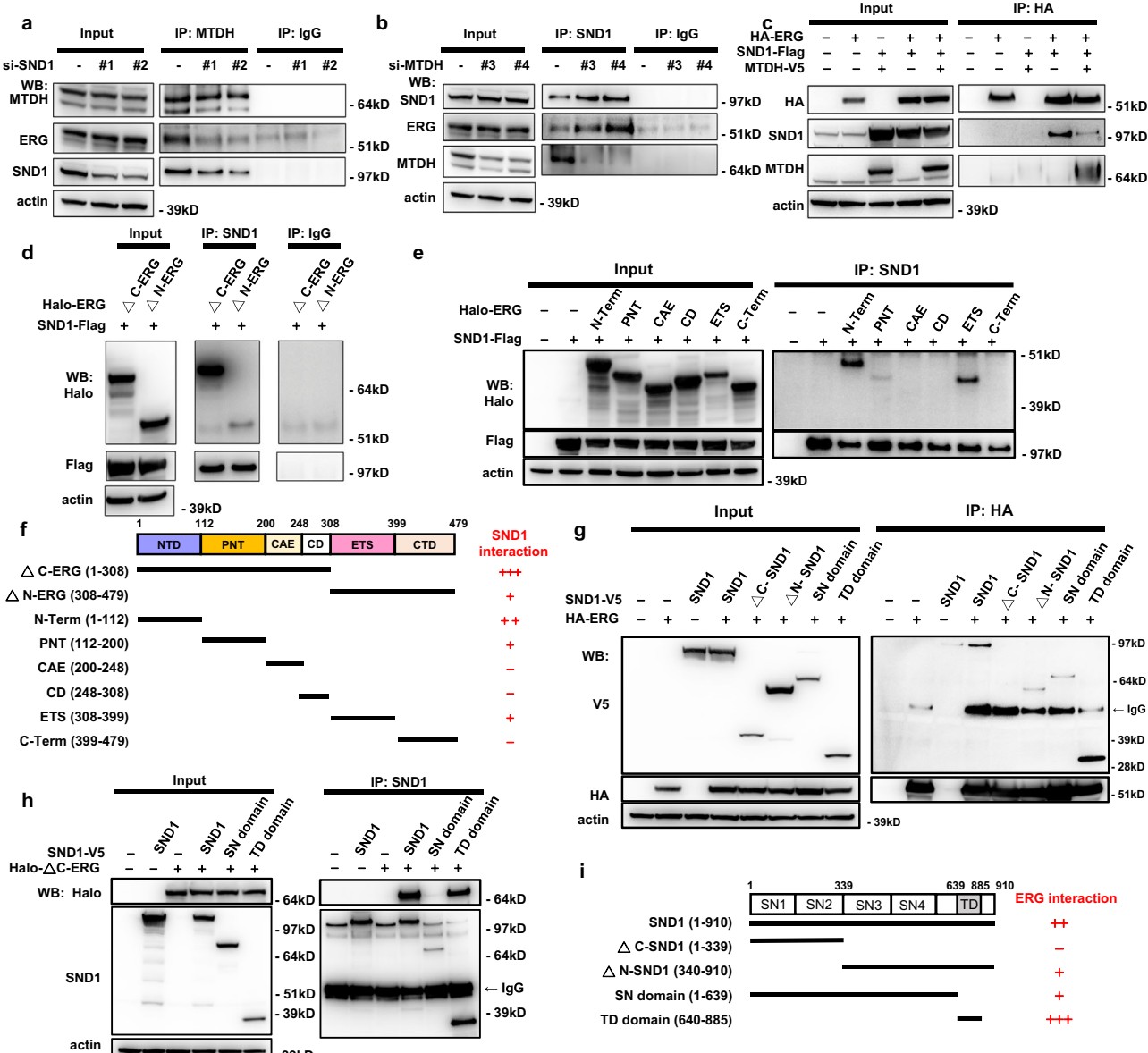

**Fig. 2 | Identification of protein domains involved in ERG-SND1/MTDH inter-actions. a** Co-IP experiment between endogenous MTDH and ERG proteins in VCaP cells transfected with non-targeting control (-) or *SND1* targeting (#1 and #2) siRNA oligos. **b** Co-IP experiment between endogenous SND1 and ERG proteins in VCaP cells transfected with non-targeting control (-) or *MTDH* targeting (#3 and #4) siRNA oligos. **c** Co-IP experiment between epitope-tagged ERG, SND1, and MTDH proteins in HEK293 cells. **d**, **e** Co-IP experiments between indicated fragments of

epitope-tagged ERG and SND1 in HEK293 cells. **f** Schematic representation of Halo-tagged ERG proteins used in (**d**, **e**) and their interaction with full-length SND1. **g**, **h** Co-IP experiments between indicated fragments of SND1-V5, HA-ERG (**g**) or HALO-ΔC-ERG (**h**) in HEK293 cells. **i** Schematic representation of V5-tagged SND1 proteins used in (**g**, **h**) and their interaction with ERG. kD, kilodalton. Experiments were repeated 2 (**g**, **h**), 3 (**c**) or 4 (**d**, **e**) times with similar results. Source data are provided as a Source data file.

gene expression, we performed RNA-Seq experiments using cells transfected with two independent siRNAs targeting either *ERG* or *SND1*, as well as non-targeting siRNA as a negative control. Comparison of transcriptional changes caused by the knockdown of *ERG* or *SND1* revealed significant overlap between both upregulated and

downregulated genes (Fig. 4a, b, Supplementary Data 2). Gene set enrichment analysis (GSEA) of these changes demonstrated highly significant pathway overlap, especially among the downregulated genes (Fig. 4c). Hallmark pathways such as "E2F gene targets" and "G2M checkpoint" showed the most significant downregulation in

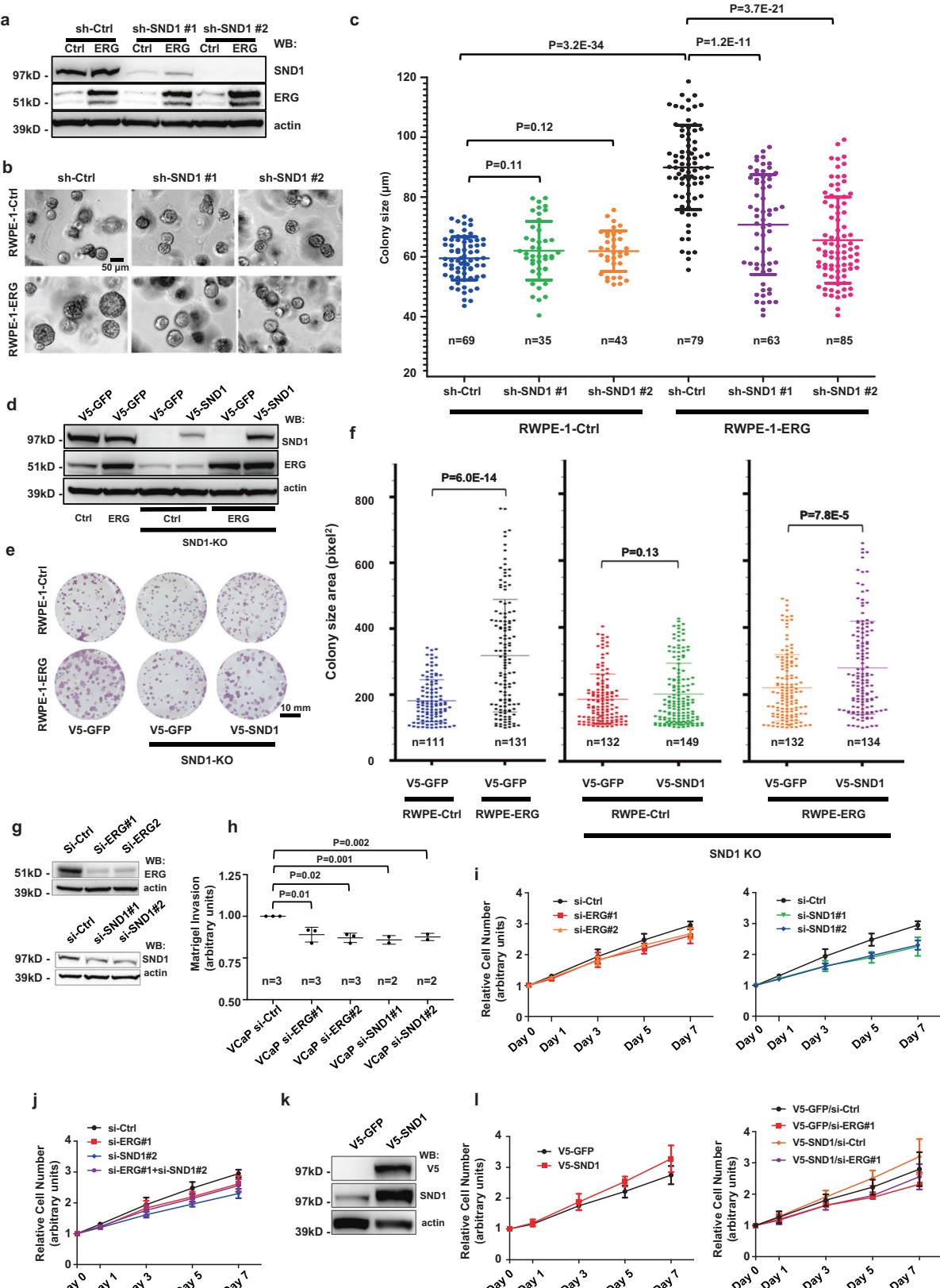

both siERG and siSND1 cells (Fig. 4d). Interestingly, while transcriptional changes were similar in siERG and siSND1 cells, they were not identical. ERG regulated a much larger number of genes than SND1, and several well-known targets of ERG in VCaP cells as *ARHGDIB, PLA1A, LAMC2*[40] were not significantly changed in siSND1 cells (Supplementary Fig. 4). ERG is also known to regulate *YAP1* of the Hippo

signaling pathway[6], but we found that the *YAP1* gene was not regulated by *SND1* (Supplementary Fig. 4). In contrast, G1/S cell cycle transition proteins Cyclin E (*CCNE1*) and its partner *CDK2* were significantly regulated by both *ERG* and *SND1* (Fig. 4e). Interestingly, the simultaneous knockdown of both *ERG* and *SND1* did not show an additive effect on the levels of *CCNE1* and *CDK2*, suggesting that *ERG* and *SND1*

**Fig. 3 | SND1 promotes proliferation of ERG-positive human prostate epithelial cells. a** Western blot analyses of RWPE-Ctrl and RWPE-ERG cells stably transduced with sh-control, sh-SND1#1 or sh-SND1#2 lentiviruses and analyzed with indicated antibodies. **b, c** Brightfield images (**b**) and colony size quantitation (**c**) of RWPE-Ctrl and RWPE-ERG colonies transduced with sh-Ctrl, sh-SND1#1 or sh-SND1#2 lentiviruses after 5 days in 3D drop culture. Colony size was determined using ImageJ. The graph shows mean +/− standard deviation (SD). Two-tailed Student's t-test. *n*, number of analyzed colonies. **d** Western blot analyses of parental and CRISPR/Cas9-generated *SND1⁻/⁻* RWPE-Ctrl and RWPE-ERG (SND1-KO) cells stably transduced with V5-GFP or V5-SND1 lentiviruses. **e, f** Brightfield images of crystal violet-stained 2D colonies (**e**) and colony size quantitation (**f**) of parental and CRISPR/Cas9-generated *SND1⁻/⁻* RWPE-Ctrl and RWPE-ERG (SND1-KO) cells stably transduced with V5-GFP or V5-SND1 lentiviruses. Cells were plated at low density and allowed to form colonies for 10 days. Colony size was determined using ImageJ. The graph shows mean +/−SD. Two-tailed Student's t-test. *n* indicates number of analyzed colonies. Scale bar in (**e**), 10 mm. **g** Western-blot analyses of VCaP cells

transfected with si-Ctrl, si-ERG#1, si-ERG#2, si-SND1#1 or si-SND1#2 siRNA oligos and analyzed with indicated antibodies. **h** Matrigel invasion assay of cells described in (**g**). Invasion data are presented in arbitrary units with values in si-Ctrl cells adjusted to 1. Data represent mean +/−SD. Combined data from independent experiments with each biological replicate (*n* = 3 for siCtrl, siERG and *n* = 2 for siSND) representing the mean of 6 technical replicates. *p*-values determined using two-tailed Student's t-test. **i, j** CellTiter-Glo assay of VCaP cells transfected with indicated siRNAs. Data represent means +/−SD. Combined data from 3 independent experiments with each biological replicate (*n* = 3) representing the mean of 6 technical replicates. **k** Western-blot analysis of VCaP cells stably transduced with V5-GFP or V5-SND1 lentiviruses and analyzed with indicated antibodies. **l** CellTiter-Glo assay of VCaP cells stably transduced with V5-GFP or V5-SND1 lentiviruses and transfected with indicated siRNA constructs. Data represent means +/−SD. Combined data from 3 independent experiments with each biological replicate (*n* = 3) representing the mean of 6 technical replicates. Source data are provided as a Source data file.

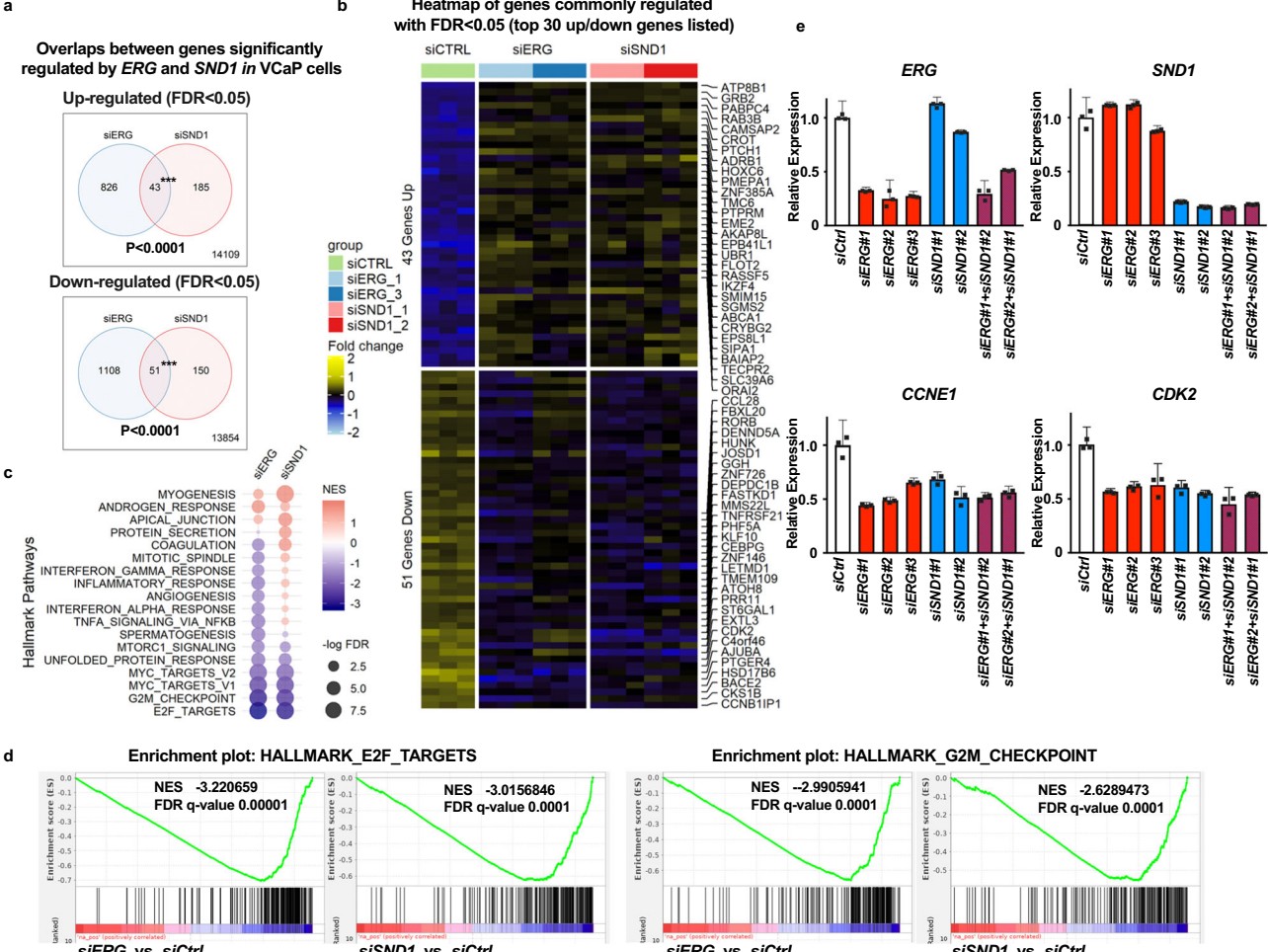

**Fig. 4 | SND1 and ERG regulate similar transcriptional programs in prostate epithelial cells. a** Venn diagrams showing common significant upregulated or down-regulated genes (FDR<.05) between siERG and siSND1 cells determined by limma analysis. Gene expression determined by RNA-Seq of VCaP cells transiently transfected with non-targeting siRNA (siCtrl, *n* = 3), siRNAs targeting *ERG* (siERG, using 2 independent siRNA oligos, *n* = 3 for each) and siRNAs targeting *SND1* (siSND1, using 2 independent siRNA oligos, *n* = 3 for each). Significance of overlap determined by two-tailed Chi-square with Yates correction. **b** Heatmap of gene expression changes of commonly regulated genes determined by RNA-seq

described in (**a**). **c** Identified by Gene Set Enrichment Analysis (GSEA) Hallmark Signaling pathways changes in siERG and siSND1 cells. **d** GSEA plots of two most significantly downregulated Hallmark pathways (E2F_targets and G2M_checkpoint) in both siERG (siERG_vs_siCtrl) and siSND1(siSND1_vs_siCtrl) VCaP cells. NES, -normalized enrichment score. FDR, -false discovery rate. **e** qRT-PCR analysis of expression of select ERG/SND1-regulated genes in VCaP transfected with indicated siRNA oligos. Gene expression data normalized using 18 s ribosomal RNA. Data represent mean +/− SD (*n* = 3 experimental replicates). Two-tailed student's t-test. Source data are provided as a Source data file.

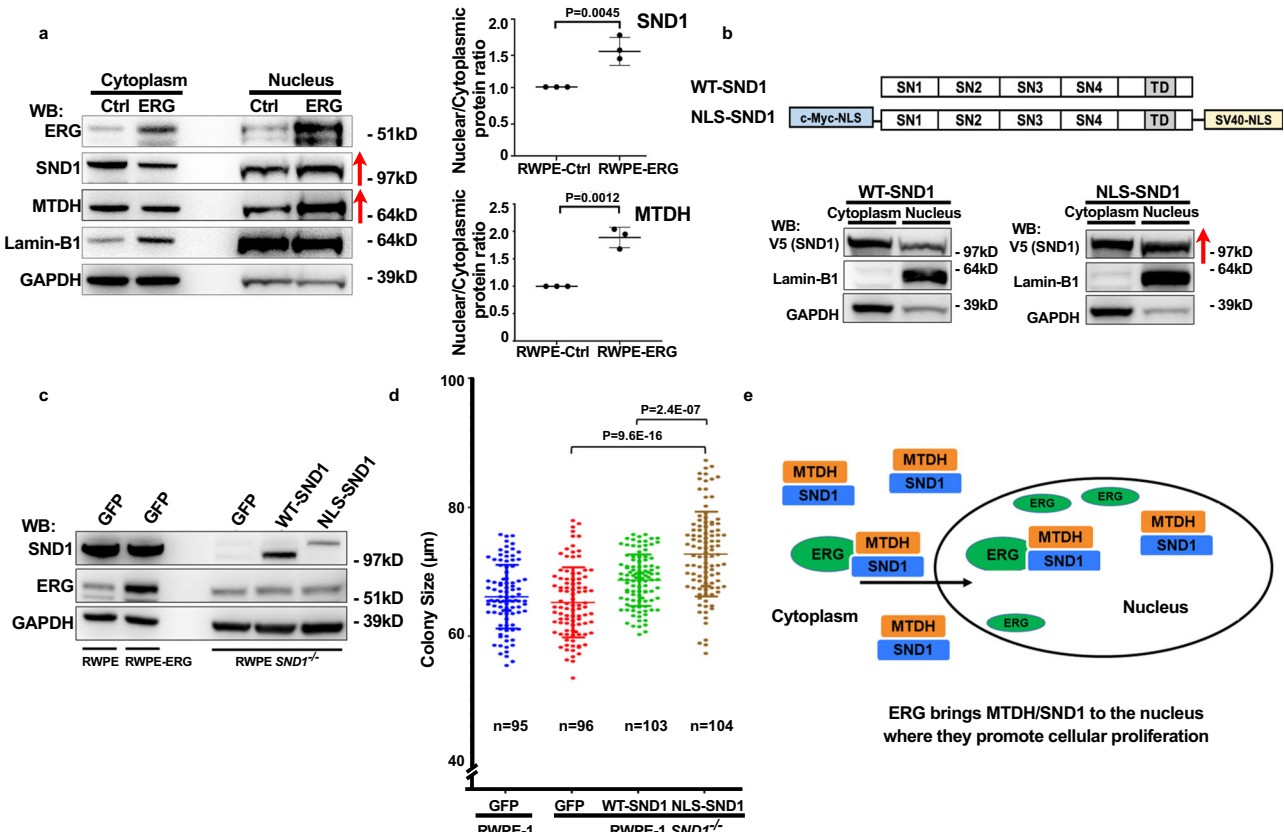

**Fig. 5 | ERG promotes nuclear localization of SND1 and nuclear SND1 stimulates the proliferation of prostate epithelial cells. a** Western blot analysis of cytosolic and nuclear proteins from RWPE-Ctrl and RWPE-ERG cells probed with indicated antibodies. The graph shows means from 3 biological replicas +/− SD with values in RWPE-Ctrl control cells adjusted to 1. *P*-values determined using two-tailed Student's t test. **b** Generation and analysis of subcellular localization of NLS-SND1. V5-SND1 control or V5-NLS-SND1 constructs were expressed in HEK293 cells, cells were fractionated and analyzed by Western blotting with indicated antibodies. **c** Western blot analysis of parental and CRISPR/Cas9-generated *SND1*−/− RWPE-1 cells stably transduced with V5-GFP, V5-SND1 or V5-NLS-SND1constructs and probed with indicated antibodies. **d** Colony size quantitation of parental and *SND1*−/− RWPE-1 cells stably transduced with V5-GFP, V5-SND1 or V5-NLS-SND1constructs. Colony size was determined using ImageJ. The graph shows mean +/− SD. Two-tailed Student's t-test. *n* indicates number of analyzed colonies. **e** Model showing the proposed mechanism of ERG in SND1-mediated activation of cell proliferation. Experiments were repeated 2 (**b**) or 4 (**d**) times with similar results. Red arrows in (**a**, **b**) indicate increase in nuclear SND1 and MTDH. Source data are provided as a Source data file.

are regulating the same signaling pathway that controls the expression of these genes (Fig. 4e).

## ERG promotes nuclear localization of SND1 and nuclear SND1 stimulates proliferation of prostate epithelial cells

While the ERG protein is primarily localized to the nucleus, the majority of SND1 is present in the cytoplasm of prostate epithelial cells (Supplementary Fig. 1). We hypothesized that ERG-SND1/MTDH interaction may increase nuclear localization of SND1/MTDH protein complex. Indeed, cytoplasmic/nuclear fractionation of RWPE-1-ERG and control RWPE-1-GFP cells revealed a significant increase in the levels of SND1/MTDH in the nucleus in ERG overexpressing cells (Fig. 5a). To determine whether nuclear-targeted SND1 is more potent than wild-type protein in stimulating proliferation of prostate epithelial cells, we generated the SND1 expression construct with exogenous nuclear localization signals (NLSs) (Fig. 5b). We then used previously generated RWPE-1-*SND1*−/− cells (Fig. 3d) to re-express either the wild-type or the NLS versions of SND1 (Fig. 5c). Despite the lower levels of expression compared to wild-type SND1, the NLS-SND1 significantly increased the size of 3D colonies of RWPE-1-*SND1*−/− cells growing in organoid culture (Fig. 5d). Thus, forced localization of SND1 to the nucleus using expression of NLS-SND1 was able to phenocopy the ERG-overexpression phenotype and stimulated the colony size of RWPE-1 cells. These data in conjunction with evidence that endogenous SND1 is necessary for the ERG-mediated increase in colony size in RWPE-1

cells (Fig. 3a–c) indicate that promotion of nuclear localization of SND1/MTDH is one of the critical functions of ERG in the transformation of prostate epithelial cells (Fig. 5e).

## Role of endogenous *Snd1* in mouse prostate gland homeostasis

To investigate the functional significance of *Snd1* in PC in vivo we utilized mouse genetics. Conventional ES cell gene targeting technology was used to generate mice with a conditional *Snd1* allele (Fig. 6a, b). *Snd1* was deleted in mouse prostate epithelium using a previously generated PB-Cre4 line[41] (Fig. 6c). As expected, conditional deletion of *Snd1 in PB-Cre4/Snd1fl/fl* males resulted in the recombination of the *Snd1* gene and produced a significant decrease in SND1 protein levels in the prostate glands (Fig. 6d, e). Analyses of resulting prostate glands with prostate-epithelium-specific ablation of *Snd1* revealed no significant changes in gross appearance, weight, and histological appearance of mutant glands (Fig. 6f–i). Thus, we concluded that *Snd1* is not necessary for normal prostate gland development or homeostasis in vivo.

## Endogenous *Snd1* promotes ERG-mediated increase in mouse prostate organoid size

Overexpression of ERG in the mouse prostate gland results in minor phenotypes and only very old (2.5–3 year old) males present with partially penetrant prostate tumors[6]. However, we noticed that *ERG* expressing mouse prostate epithelial cells growing as organoids

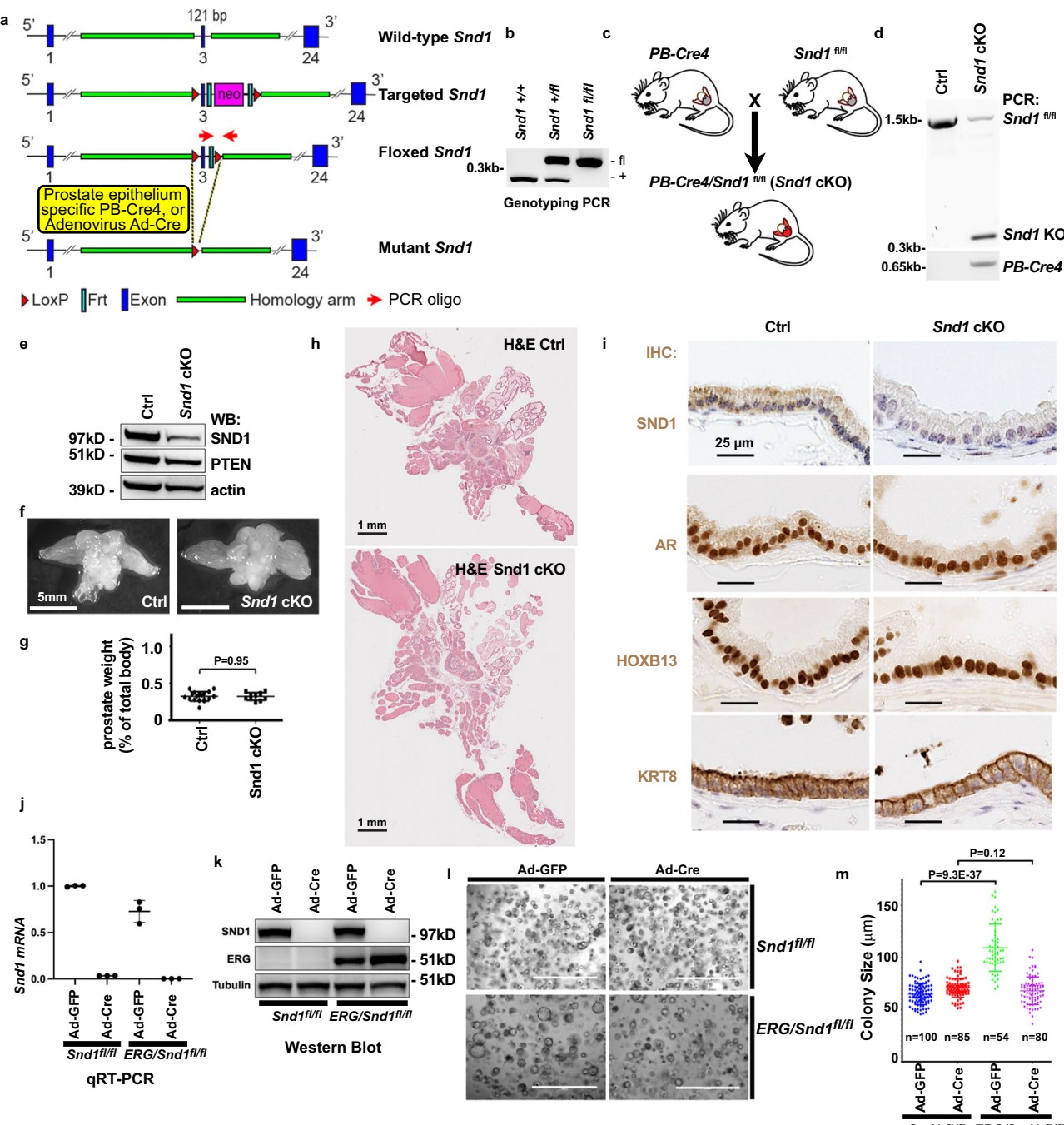

**Fig. 6 | Role of endogenous *Snd1* in mouse prostate gland homeostasis and growth of primary mouse prostate organoids in culture. a** Generation of conditional allele and mice with prostate-specific knockout of *Snd1*. Schematic representation of the wild-type, targeted, conditional (floxed), and Cre-recombined (mutant) alleles of *Snd1*. **b** PCR genotyping of *Snd1* allele from wild-type (*Snd1^+/+*), heterozygous (*Snd1^+/flox*), and homozygous (*Snd1^flox/flox*) mice. Note the absence of wild-type *Snd1* PCR allele in *Snd1^flox/flox* sample. **c** Generation of prostate-specific *Snd1* knockout mice. **d** PCR genotyping of *Snd1* allele from whole prostate glands of wild-type (Ctrl) and *PB-Cre4/Snd1^flox/flox* (*Snd1* cKO) mice. Note the appearance of recombined knockout (*Snd1* KO) allele in *Snd1* cKO sample. Small amount of remaining *Snd1^flox/flox* allele is due to the prostate stromal cell population, which is not targeted by *PB-Cre4*. **e–h** Western blot analysis (**e**), gross appearance (**f**), relative weights (**g**) and histological analysis (**h**) of prostate glands from 1-year-old control

(Ctrl) and *PB-Cre4/Snd1^flox/flox* (*Snd1* cKO) mice. Graph shows means +/− SD. *n* = 18 for control mice. *n* = 10 for *Snd1* cKO mice. Two-tailed Student's t-test. **i** Immunohistochemical (IHC) staining of ventral prostate gland sections from 1-year-old Ctrl and *Snd1* cKO mice with indicated antibodies. **j, k** RT-PCR (**j**) and Western blot analysis (**k**) of primary mouse prostate epithelial cells isolated from *Snd1^flox/flox* and ERG/*Snd1^flox/flox* mice and transduced with adenovirus carrying Cre (Ad-Cre) or GFP control (Ad-GFP). The graph shows mean of 3 samples +/− SD. Two-tailed Student's t-test. **l, m** Brightfield images (**l**) and colony size quantitation (**m**) of cells described in (**j, k**) cultured for 5 days in 3D organoid drop culture system. Colony size was determined using ImageJ. The graph shows mean +/− SD. Two-tailed Student's t-test. *n* indicates number of analyzed colonies. Experiment was repeated 4 times with similar results. Scale bar, 1 mm. Source data are provided as a Source data file.

displayed an increase in average colony size, which is similar to the *ERG*-overexpression phenotype in human prostate epithelial cells. To analyze the potential role of *Snd1* in this ERG-mediated phenotype, we isolated *ERG*-overexpressing and control primary prostate epithelial cells from mice with conditional *Snd1* and deleted *Snd1* using Adenovirus-Cre infection ex vivo (Fig. 6j, k). While *Snd1* wild-type prostate organoids displayed ERG-mediated increase in colony size, the difference in size between control and ERG-expressing cells was completely eliminated upon ablation of *Snd1* (Fig. 6l, m). Therefore, we conclude that similar to human prostate epithelial cells, *Snd1* is an important contributor to ERG-mediated promotion of organoid growth in mouse prostate epithelial cells.

## Ablation of endogenous *Snd1* negatively impacts PC growth in vivo

To investigate the role of *Snd1* in the context of autochthonous PC, we utilized *PB-Cre4/Pten^fl/fl^/ERG* mice that develop high-grade prostatic intraepithelial neoplasia and prostate adenocarcinoma as young adults (8–12 months after birth). We generated cohorts of *PB-Cre4/Pten^fl/fl^/ERG* and *PB-Cre4/Pten^fl/fl^/ERG/Snd1^fl/fl^* mice and analyzed them at 1 year after birth (Fig. 7a–c). Ablation of *Snd1* had a prominent negative impact on prostate tumor development and growth (Fig. 7d–f, Supplementary Fig. 5). The weights of the prostate glands and the incidence of prostate carcinoma were significantly decreased in *PB-Cre4/Pten^fl/fl^/ERG/Snd1^fl/fl^* males (Fig. 7d, e, Supplementary Fig. 6). While the overall size of the stromal prostate gland compartment was not affected, the epithelial compartment was reduced in *Snd1* mutants (Fig. 7f, Supplementary Fig. 5). The remaining *Snd1^−/−^* epithelial cells continued to express luminal epithelial cell markers KRT8, HOXB13, and AR (Supplementary Fig. 7). Analyses of proliferation and apoptotic cell death revealed significant reduction in proliferation and increased apoptosis in the prostate epithelial cell compartment of *PB-Cre4/Pten^fl/fl^/ERG/Snd1^fl/fl^* mice (Fig. 7g). We concluded that while the ablation of endogenous *Snd1* does not significantly impact normal prostate gland homeostasis, it negatively regulates the expansion and growth of autochthonous PC.

## SND1 is necessary for ERG-mediated regulation of imprinted genes

To determine the transcriptional impact of *Snd1* ablation in mouse prostate in vivo, we performed RNA-Seq analysis on prostate glands from *PB-Cre4/Pten^fl/fl^/ERG* and *PB-Cre4/Pten^fl/fl^/ERG/Snd1^fl/fl^* mice (Fig. 8a, Supplementary Data 3). For this analysis, we utilized young males soon after activation of Cre and ERG expression and before the histological differences between prostate glands in these genotypes become apparent. Inactivation of *Snd1* resulted in significant changes in gene expression. Interestingly, we noticed that many of *Snd1* cKO-downregulated genes are the genes that were upregulated by ERG (Fig. 8a, Supplementary Data 3). Therefore, *Snd1* ablation in vivo attenuated the expression of a subset of ERG target genes. Analysis of the most substantial changes (FDR < 0.05 and FC > 2) revealed significant overlaps between both upregulated and downregulated genes (Fig. 8b, c, Supplementary Data 3). Remarkably, 7 of 68 overlapping genes that were upregulated in vivo by overexpression of *ERG* and then downregulated by ablation of *Snd1* are genes known to be imprinted in the mouse (*Igf2, Dlk1, Rtl1, MirG, Rian, Ppbp, AF357425*)[42] (Fig. 8c, d, Supplementary Fig. 8) While many imprinted genes are known to drive tissue growth, the significance of ERG-mediated upregulation of imprinted genes is presently not clear. Overall, we conclude that endogenous *Snd1* is necessary for a part of the ERG-regulated transcriptional program and notably influences the expression of a subset of imprinted genes.

## Discussion

In this study we found that ERG interacts with the MTDH/SND1 protein complex and overexpression of endogenous ERG in prostate epithelial cells increases the nuclear localization of MTDH/SND1 and promotes the growth of prostate cells in vitro and in vivo. Functional loss-of-function and gain-of-function experiments demonstrated that this is a previously unknown mechanism of ERG-mediated transformation. These findings also highlight an important role of SND1 in PC. While the overexpression of MTDH/SND1 in human PC is well documented in the literature, the functionally important role of these proteins has been mostly demonstrated using cancer cell lines[43,44]. The role of endogenous *Mtdh* in autochthonous PC has been investigated using germline deletion of *Mtdh* in mouse model of cancer driven by prostate-specific overexpression of SV40 large and small tumor antigens (TRAMP model)[45]. Double mutant mice displayed prolonged tumor latency, reduced tumor burden and a reduction in metastasis[45]. Studies in breast cancer models revealed that the primary function of MTDH in cancer involves interaction with SND1 and protection from degradation[30,31,33]. These important findings highlighted the critical role of SND1, but the role of endogenous *Snd1* was not extensively investigated. In this study, we generated mice with conditional allele of *Snd1* and used them to analyze the role of *Snd1* in PC. This approach revealed an important role of endogenous *Snd1* for the development of autochthonous PC.

We choose to utilize a mouse PC model driven by overexpression of *ERG* combined with homozygous loss of *Pten*. This model is highly relevant clinically because overexpression of *ERG* strongly correlates with loss of *PTEN* in human PC[46]. In our model, the expression of *ERG* is driven by a highly expressed transgene integrated into an intergenic region on mouse chromosome 1 and activated by androgen signaling in mouse prostate epithelial cells[6,47]. This is similar to the situation in the human prostate gland that occurs after a genetic recombination between the strong androgen-driven promoter of *TMPRSS2* and the coding region of *ERG*[1]. A recently generated mouse model that mimics this genetic recombination event has also been used in PC research[16,48]; however, we decided against using this model because unlike human *TMPRSS2*, mouse *Tmprss2* is not androgen driven[49–51]. In addition, single cell RNA sequencing reveled that in the mouse prostate gland *Tmprss2* is expressed at 13–23 fold lower levels than human *TMPRSS2*, when normalized to expression of Keratin8/Keratin18 genes, respectively[52]. Similarly, another widely used model of *ERG* expression[53] utilizes endogenous *Gt(ROSA)26Sor* promoter that is also not androgen driven and shows ubiquitous but low level of expression[54], which in the mouse prostate gland is relatively similar to the expression level of *Tmprss2*[52]. Human *TMPRSS2-ERG* recombination results in very high levels of androgen driven expression of *ERG*[1] and this is likely to be important to mimic in mouse models of PC.

In our *PB-Cre4/Pten^fl/fl^/ERG/Snd1^fl/fl^* model of PC the cancer-driving *Pten* inactivation co-occurs with deletion of *Snd1*, as both of these events are carried out by same Cre recombinase. Therefore, this model can only inform on the role of *Snd1* in cancer initiation and not cancer maintenance and progression. It will be important in the future to use additional models that can ablate *Snd1* in established tumors, as these models will be more informative regarding potential utility of SND1 inactivation/inhibition in the treatment of PC patients. In addition, it will be interesting to analyze the role of SND1 in PC that does not display overexpression of ERG. While ERG promotes SND1 function, the presence of endogenous SND1 may be important for both ERG-positive and ERG-negative tumors. We found that *Snd1* is not required for normal prostate gland homeostasis; however, normal prostate epithelium exhibits very low proliferation and the lack of an observable role for *Snd1* may be due to the slow rates of normal prostate gland self-renewal.

We found that ERG increases the nuclear localization of SND1, and nuclear localized SND1 mutant is a much more potent driver of cellular proliferation of prostate epithelial cells than the wild-type protein. The mechanisms responsible for these findings are presently unknown. SND1 is a multifunctional protein which has been implicated in the

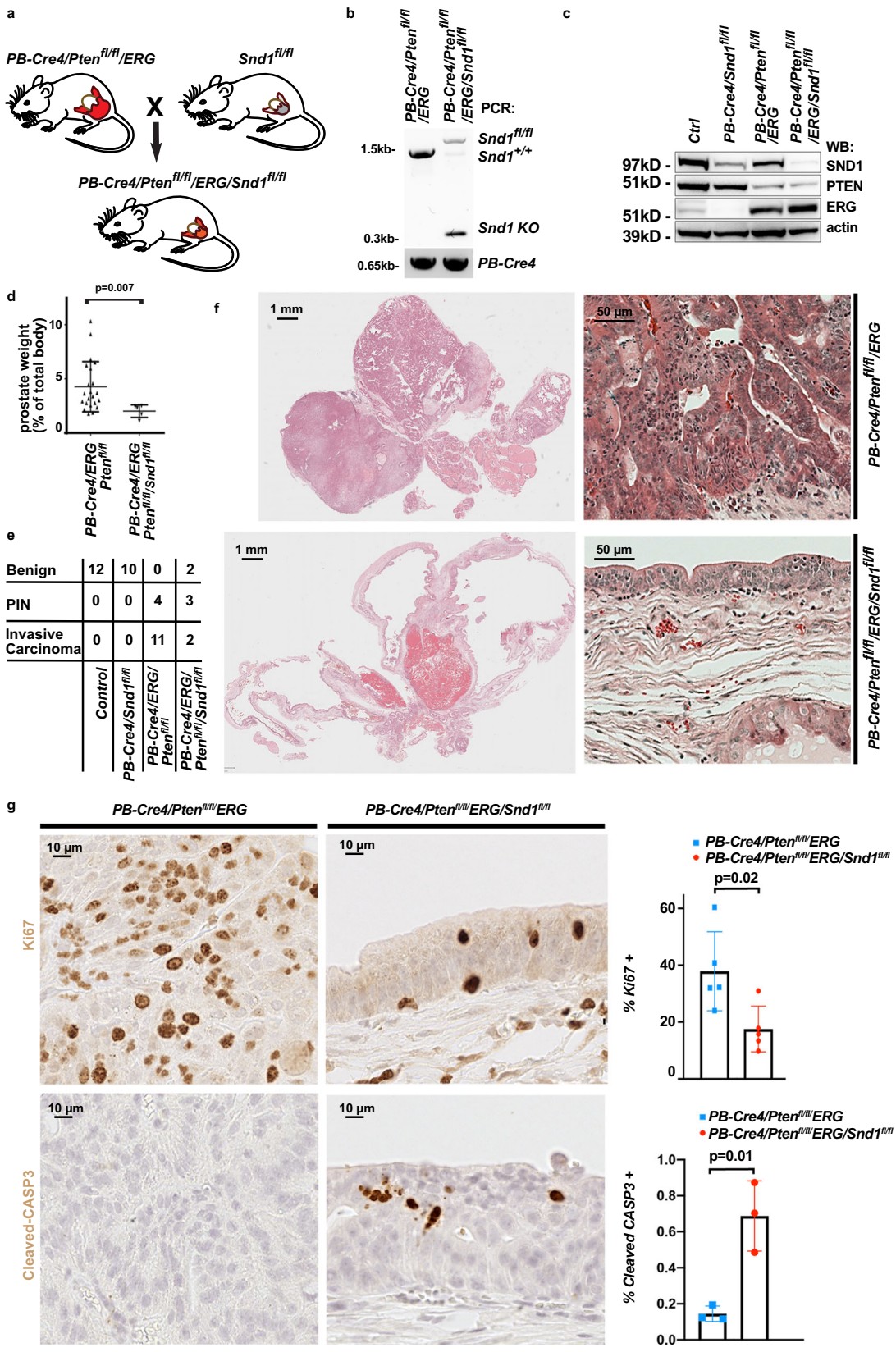

regulation of transcriptional activation, alternative splicing, ubiquitination, mRNA stabilization and RNA interference[34,37,39,55–58]. Many of these functions can be caried out in the nucleus. In this study we identified a significant impact of SND1 on the regulation of mRNA

levels; however, future studies will have to assess the significance of these changes with respect to tumor promotion. We also attempted to analyze whether ERG impacts SND1 chromatin localization. However, our Chip-Seq experiments with various commercially available anti-

**Fig. 7 | *Snd1* is necessary for tumor growth in an autochthonous mouse model of prostate cancer. a** Generation of control *PB-Cre4/Pten^flox/flox^/ERG* and *Snd1*-mutant *PB-Cre4/Pten^flox/flox^/ERG/Snd1^flox/flox^* mice. **b** Representative image of typical PCR genotyping of *Snd1* allele from prostate glands of mice with indicated genotypes. Note the appearance of recombined knockout (*Snd1* KO) allele in *Snd1* cKO sample. Small amount of remaining *Snd1^flox/flox^* allele is due to the prostate stromal cell population, which is not targeted by *PB-Cre4*. **c** Western blot analysis of ventral prostate glands from 1-year-old mice with indicated genotypes using indicated antibodies. **d** Relative weights of mouse prostate glands from 1-year-old mice with indicated genotypes. Graph shows means +/− SD. *n* = 22 for *PB-Cre4/Pten^flox/flox^/ERG* mice. *n* = 7 for *PB-Cre4/Pten^flox/flox^/ERG/Snd1^flox/flox^* mice. Two-tailed Student's t-test.

**e** Pathological evaluation of prostate gland histology from 1-year-old mice with indicated genotypes. Table shows number of individual mice with indicated diagnosis. PIN, prostatic intraepithelial neoplasia, a precancerous prostate gland lesion. **f** Haematoxilin & eosin (H&E) staining of prostate glands from 1-year-old mice with indicated genotypes. *n* = 7 for each genotype. See also Supplementary Fig. 5. **g** Immunohistochemical (IHC) staining and quantitation of prostate gland sections from *PB-Cre4/Pten^flox/flox^/ERG* and *PB-Cre4/Pten^flox/flox^/ERG/Snd1^flox/flox^* mice with indicated antibodies. Data represent mean +/− SD. *n* = Two-tailed Student's t-test. *n* = 5 individual mice for Ki67. *n* = 3 individual mice for cleaved Caspase3. Source data are provided as a Source data file.

SND1 antibodies were unsuccessful. Future experiment will need to address whether ERG is only involved in nuclear retention of SND1, or whether both proteins continue to function together as a complex in mediating the growth promoting function of SND1 in the nucleus of prostate epithelial cells.

We report here that overexpression of *ERG* in mouse prostate gland results in prominent upregulation of many imprinted genes and the deletion of *Snd1* downregulates their expression indicating that they are at least partially driven by SND1. While these findings are intriguing, the significance of these changes is presently unknown. Many of these imprinted genes play important growth regulatory functions and may be potentially responsible for growth promoting function of ERG and SND1. However, we did not identify a connection between ERG and the regulation of imprinted gene expression in cells in culture. Moreover, the prominent differences in imprinted gene expression that are seen in vivo disappear withing a few days after primary cells are placed in culture. The mechanisms responsible for ERG-mediated increases in the levels of imprinted genes are presently unknown. Our preliminary analysis of DNA methylation in imprinting control regions (ICRs) of these genes did not reveal differences between the prostate glands from ERG-expressing and control mice. The significance of broad imprinted gene upregulation in ERG-mediated prostate cancer will have to be investigated in the future studies.

While ERG has been implicated in regulation of variety of cancer-relevant signal transduction pathways, the exact mechanism that is responsible for its role in PC initiation is still unknown and it will likely remain the subject of active investigation. The ERG-MTDH/SND1 interaction uncovered in this study appears to play an important role in this process.

## Methods
### Animal models
All procedures involving mice and experimental protocols were approved and performed in accordance with guidelines from the Institutional Animal Care and Use Committee (IACUC) of Fred Hutchinson Cancer Center (FHCC) and followed NIH ethical guidelines for animal welfare. The maximal tumor size/burden permitted by our IACUC is 2.0 cm in diameter and it was not exceeded in this study. Only male mice were used in prostate gland analysis. Mice with a conditional *Snd1* allele containing exon 3 flanked by LoxP sequences were generated by TIGM using conventional embryonic stem cell gene targeting technology. PCR with oligos Snd1-F (5′-GGAACTGTTGCTGTGTTCGT-3′) and Snd1-R (5′-GCTAAAGAGTCCCTAGAAAG-3′) was used for genotyping (wild-type allele, 236 base pairs (bp); floxed allele, 364 bp). *PB-Cre4/Snd1^fl/fl^* mice were generated by crossing *PB-Cre4*[41] with *Snd1^fl/fl^* mice. *PB-Cre4/Pten^fl/fl^/ERG* mice were generated by crossing *PB-Cre4/Pten^fl/fl^* mice[51] with Tg(Pbsn-ERG)1Vv mice[47]. *PB-Cre4/Pten^fl/fl^/ERG/Snd1^fl/fl^* were made by crossing *PB-Cre4/Pten^fl/fl^/ERG* mice with *Snd1^fl/fl^* mice. All mice were maintained on a mixed 129S1/SvlmJ/C57BL/6J/genetic background. *Snd1^fl/fl^* mice will be available from MMRRC (RRID: MMRRC_069917-UCD).

### Cell line culture, transient transfection, and lentivirus production
ERG-positive human tumorigenic prostate cancer cell line VCaP, immortalized nontumorigenic prostate epithelial cell line RWPE-1, and human embryonic kidney 293T (HEK293T) cells were purchased from ATCC. LuCaP 35CR cells derived from ERG-positive patient-derived prostate cancer xenografts were obtained from Dr. Peter Nelson (FHCC). VCaP and HEK293T cells were cultured in DMEM media (Thermo Fisher, 11965-092) with 10% fetal bovine serum (Hyclone), sodium pyruvate (Thermo Fisher, 11360-070), non-essential amino acids (Thermo Fisher, 11140-050) and primocin (InvivoGen, ant-pm-2). LuCaP 35CR cells were cultured in DMEM media with 10% fetal bovine serum and 1% penicillin/streptomycin (Thermo Fisher, 15140-122). RWPE-1 cells were cultured in K-SFM keratinocyte medium (Thermo Fisher, 17005-042) with 1% fetal bovine serum and 1% penicillin/streptomycin. RWPE-Ctrl and RWPE-ERG cells were previously described[6]. Transient transfection with siRNA oligos was performed using Lipofectamine RNAiMAX according to manufacturer protocols (Thermo Fisher, 13778-075). Transient transfection with plasmid DNA was performed using polyethylenimine as previously described[59]. Lentiviruses were produced in HEK293T cells as described[60]. Cells stably transduced with lentiviruses were selected with puromycin (Sigma, P8833) or blasticidin (Thermo Fisher, A11139-03). Human cells with stable knockdown of *SND1* were generated using pGIPZ lentiviruses. RWPE-Ctrl and RWPE-ERG cells with knockout of endogenous *SND1* were generated using Crispr/Cas9 technology by stable lentiviral transduction with sgRNAs, selection, and transient transduction with adenovirus carrying Cas9 (Ad5CMVspCas9) or GFP (Ad5CMVeGFP), as control, which were purchased from the University of Iowa Viral Vector Core Facility. *SND1* knockout single-cell clones were isolated into 96-well plates by sorting RFP-positive cells using FACS (Sony SH800S). Knockout and knockdown efficiencies were validated by Western blot analysis. The identity of cell lines was confirmed by short tandem repeat analysis. All cell lines were tested negative for mycoplasma contamination using MycoProbe Mycoplasma Detection Kit.

### Primary mouse prostate cell matrigel drop culture
Mouse prostate ventral lobe tissue was dissected from 5-month-old wild-type, *ERG*, *Snd1^fl/fl^*, and *ERG/Snd1^fl/fl^* mice. Prostate tissues were then minced, transferred to 15 ml tube and digested with 5 mg/ml collagenase in Advanced DMEM/F-12 medium for 1 h on a shaker at 37 °C (Thermo Fisher, 12634-010). Tissue pieces were washed and then dissociated to single cells by digesting with TrypLE™ Express Enzyme (Thermo Fisher, 12605-010) containing 10 mM HEPES (Millipore, TMS-003-C) and 10 µM Y-27632 (Selleck Chemicals, S1049) at 37 °C for 15 min. Tissue was dissociated by vigorous pipetting and single cells were isolated by passing through 70 µm strainer and centrifugation. Cells were cultured using matrigel drop culture system developed by Dr. John Lee (FHCC). Cell pellet was resuspended in ice-cold matrigel (Corning, 354230) at concentration of $5 \times 10^3$ cells per 15 µl matrigel and then seeded as 15 µl drops in the middle of each well of 48-well plate. Plate was placed upside down in the $CO_2$ incubator at 37 °C for

 11

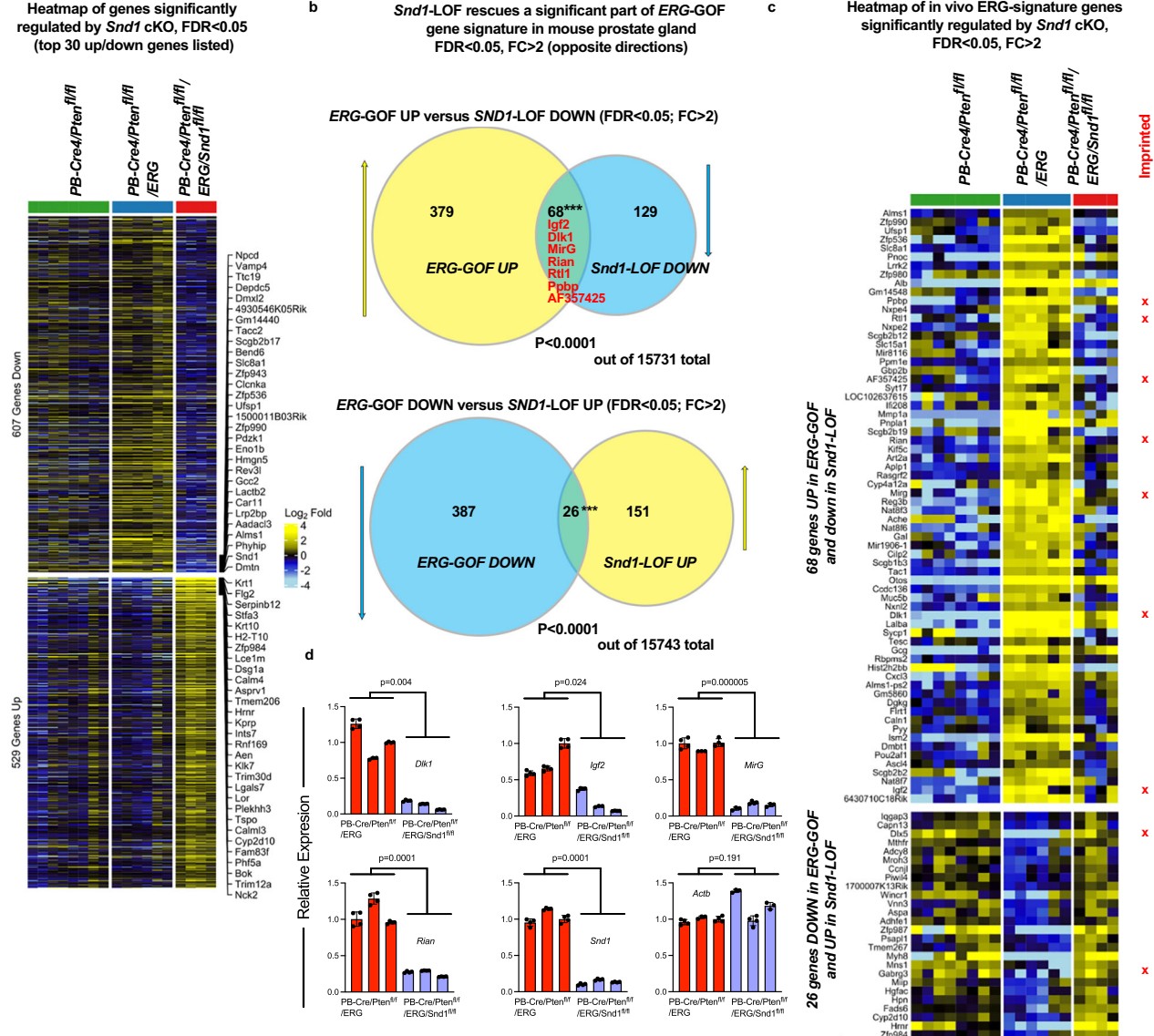

**Fig. 8 | Endogenous *Snd1* is necessary for ERG-mediated regulation of imprinted genes. a** Heatmap of gene expression changes in ventral prostate glands of mice with indicated genotypes. Each column represents an individual sequenced sample. Heatmap shows the genes that are significantly changing upon deletion of *Snd1* in ERG/*Pten* cKO prostate cancer model. Note that many genes downregulated upon ablation of *Snd1* were upregulated upon expression of ERG in *Pten* cKO prostate cancer model. **b** Venn diagrams showing significant overlaps between genes commonly regulated by *ERG* and *Snd1*. Upper diagram is showing genes that are upregulated by ERG (ERG-GOF-UP, comparison between *PB-Cre4/Pten^flox/flox^/ERG* and *PB-Cre4/Pten^flox/flox^*, FDR < 0.05 and fold change > 2) and genes that are prominently downregulated by ablation of *Snd1* (Snd1-LOF-DOWN, comparison between *PB-Cre4/Pten^flox/flox^/ERG/Snd1^flox/flox^* and *PB-Cre4/Pten^flox/flox^/ERG*, FDR < 0.05 and fold change > 2). Commonly regulated imprinted genes are listed in red. Lower diagram

is showing overlap between genes that are down-regulated by expression of ERG (ERG-GOF-DOWN) and upregulated by ablation of *Snd1* (Snd1-LOF-UP). ***$p < 0.001$. Significance was determined by two two-tailed Chi-square test with Yates correction. **c** Heatmap of gene expression changes in ventral prostate glands of mice with indicated genotypes. Heatmap shows the expression of genes overlapping on Venn diagrams in (**b**). Red X marks known mouse imprinted genes[42]. **d** qRT-PCR analysis of expression of imprinted genes *Dlk1, Igf2, MirG, Rian,* and relevant controls in ventral prostate glands of three ERG/*Pten* cKO and three ERG/*Pten*/*Snd1* cKO mice. Gene expression data are normalized using 18s ribosomal RNA. Graph shows mean +/−SD of technical replicates for $n = 3$ biologic replicate samples with significance determined between biologic replicate means using two-tailed Student's t-test. Source data are provided as a Source data file.

30 min to allow the matrigel to solidify. The plate was then inverted and 150 μl of warm organoid culture medium was added to the wells containing solidified drops of matrigel. Organoid culture medium modified from[61] contained: Advanced DMEM/F-12 media (Thermo Fisher, 12634-010), 1x B27 Supplement (Thermo Fisher, 17504-001), 10 mM HEPES, 2 mM GlutaMAX (Thermo Fisher, 35050-061), 1.25 mM N-Acetyl-L-cysteine (Sigma, A9165), 1 μM prostaglandin E2 (Tocris, 2296), 1 nM dihydrotestosterone (Sigma, D-073-1ML), 10 mM Nicotinamide, 100 μg/ml primocin, 50 ng/ml EGF (Peprotech, 315-09), 500 nM A83-01 (Tocris, 2939), 500 ng/ml R-Spondin (Peprotech,

120-38) and 100 ng/ml Noggin (Peprotech, 120-10C). Media was replaced every 2 days. To passage, the drops were incubated with 150 μl of dispase (Stem Cell Technologies, 07913) for 15 min at 37 °C, disrupted by pipetting and transferred to 15 mL conical tube. Cells were spinned down and dissociated by using 0.25% Trypsin/EDTA for 5 min at 37 °C. Trypsin was inactivated by adding DMEM/F-12 with 10% FBS and resulting cells were washed using organoid culture medium. To generate paired *Snd1^−/−^* and control primary mouse cultures, cells from *Snd1^fl/fl^* and *ERG/Snd1^fl/fl^* were infected with adenoviruses carrying Cre (Ad5CMVCre-eGFP) or GFP as control (Ad5CMVeGFP), which were

purchased from the University of Iowa Viral Vector Core Facility. Knockout efficiency was validated by PCR and western blot analysis. RWPE-1 cells were cultured in matrigel drop cultures using K-SFM medium as described above.

## Cell growth, invasion assays

For the cell proliferation assay, VCaP ($2.5 \times 10^4$) cells were plated in 96-well plates and fed every 2 days. Cell numbers were monitored by CellTiter-Glo® 2.0 Cell Viability Assay (Promega, G9242) using BioTek Synergy H1 Hybrid Multi-Mode Reader. For RWPE-1 cells growing in 3D organoid cell system, $2 \times 10^3$ cells were plated in six replicates on each well of matrigel coated 48-well plates and cultured for 5 days. The resulting colonies were imaged, and colony diameters were determined using imageJ software. For colony formation assays, 300 cells were plated on each well of six-well plate and cultured for 10 days. The resulting colonies were stained with 1xPBS containing 4% formaldehyde and 0.5% crystal violet. Each cell type was analyzed in triplicate. Colonies were imaged and colony diameters were determined using imageJ software. Cellular invasion was measured using the Cultrex 96-well Cell Invasion Assay (Trevigen, #3481-096-K) according to the manufacturer protocols. $5 \times 10^4$ cells were plated into the top chamber and invasion was measured 24 h after plating.

## Immunoprecipitation, Western blot analysis, and cytoplasmic/nuclear protein fractionation

Description of all antibodies can be found in Supplementary Table 1. For western blot analysis, cells were lysed on ice using RIPA buffer in the presence of a protease (Thermo Fisher, A32955) and phosphatase inhibitor (Thermo Fisher, A32957) cocktails. Protein extracts containing equal amounts of protein (50 µg) were solubilized in 1xLDS Sample Buffer (Thermo Fisher, NP0007) and separated by sodium dodecyl sulfate/polyacrylamide gel electrophoresis (SDS-PAGE), and transferred to PVDF membranes (Millipore, IPVH00010). Membranes were incubated with primary and species-specific HRP-labeled secondary antibodies (Jackson ImmunoResearch Laboratories), that were detected using immobilon western chemiluminescent HRP substrate (Millipore, WBKLS0500). Blots were imaged using ChemiDoc MP system (BioRad, 12003154).

For the immunoprecipitation (IP) assay, cells were lysed in IP buffer containing 50 mM Tris-HCl, pH 7.5, 1% Triton X-100, 50 mM NaCl, 0.1 mM EDTA, protease and a phosphatase inhibitor cocktails. Lysates was pre-cleaned with 30 µl Protein A/G Agarose (Thermo Fisher Scientific, 20421) for 1 h at 4 °C. Supernatant was then transferred to a new tube and incubated with anti-IgG (control), or indicated primary antibodies for 1 h followed by a 3 h incubation with 50 µl protein A/G–Sepharose beads at 4 °C. For IPs of tagged proteins, protein lysates were incubated for 4 h at 4 °C with 50 µl Anti-HA agarose beads (Thermo Fisher Scientific, 26181). Beads were then washed four times with IP buffer. Proteins were eluted using 1x LDS Sample Buffer, separated by SDS-PAGE, and then analyzed by Western blotting.

The cell fractionation assay was performed using an NE-PER™ kit (Thermo Fisher Scientific, 78833) according to the manufacturer protocols.

## Proximity ligation assay

The proximity ligation assay was performed using Duolink Proximity Ligation Assay kit according to the manufacturer protocols. Briefly, VCaP and RWPE-1 cells grown on glass coverslips were fixed with 4% Paraformaldehyde at room temperature (RT) for 15 min. Cells were permeabilized with 0.1% Triton X-100 for 10–15 min and then incubated overnight at 4 °C with anti-ERG together with or without anti-SND1 or anti-MTDH antibodies. PLUS and MINUS secondary PLA probes against rabbit and mouse IgG (Sigma, DUO92005; DUO92001) were added, and the cells were incubated at 37 °C for 1 h with gentle

agitation, followed by incubation with the ligation mix for 30 min at 37 °C. Amplification mix was then applied for 100 min at 37 °C. The coverslips were mounted on microscope slides with Mounting Medium with DAPI (Abcam, ab104139), and the cells were imaged using confocal laser scanning microscope LSM 800 (Zeiss).

## Immunofluorescent staining

VCaP cells grown on glass coverslips were fixed with 4% paraformaldehyde for 15 min. Cells were permeabilized with 1xPBS, 0.1% Triton X-100 for 15 min and incubated with anti-ERG and/or anti-SND1 and/or anti-MTDH antibodies overnight at 4 °C. Primary antibodies were detected using secondary antibodies from Jackson ImmunoResearch or Invitrogen (Supplementary Table 1). The coverslips were mounted on microscope slides using Mounting Medium with DAPI (Abcam, ab104139) and imaged using confocal laser scanning microscope LSM 800 (Zeiss).

## RNA extraction, reverse transcription-PCR (RT-PCR) and quantitative RT-PCR (qRT-PCR)

Total RNA was extracted with QIAzol (QIAGEN) followed by RNase-free DNase treatment (QIAGEN) and purification using RNeasy kit (QIAGEN). Complementary DNA was prepared with a SuperScript III First-Strand Synthesis kit (Thermo Fisher, 18080-051). Genes of interest were analyzed by RT-qPCR using a Power SYBR Green Master Mix (Thermo Fisher, 4367659) and performed by QuantStudio 6 Flex Real-Time PCR System (Applied Biosystems).

## Plasmids and siRNA oligos

All oligos used for cloning can be found in Supplementary Table 2. Retroviral plasmid encoding ΔN-ERG-ires-GFP was previously described[6]. HA-tagged ΔN-ERG cDNAs were generated by PCR using oligos HA-ERG-F and HA-ERG-R (for N-terminal HA-ERG) and ERG-HA-F and ERG-HA-R (for C-terminal ERG-HA). Resulting DNA fragments were TA cloned into pCR8/GW/TOPO Gateway entry vector using pCR™8/GW/TOPO™ TA Cloning Kit (Thermo Fisher, K250020). Lentiviral HA-ΔN-ERG and ΔN-ERG-HA expression constructs were generated using Gateway LR cloning into pLenti6/V5-DEST vector. Lentiviral N-terminal and C-terminal VA (3xFLAG-2xTEV-6xHIS-StrepIII-Beacon) tagged ERG was generated by Gateway cloning of ERG into pLD-puro-CnVA and pLD-puro-CcVA plasmids[62] (Addgene, #24587 and 24588). pFN19A plasmids containing Halo-tagged ERG fragments were gifted by Dr. Arul M. Chinnaiyan (University of Michigan)[13]. These ERG fragments were subcloned into pFN21A using Flexi® Cloning System (Promega, R1851). Human Myc-Flag-tagged full-length SND1 plasmid was purchased from (OriGene, RC200059). Full-length, truncated and NLS-tagged SND1 cDNAs were generated by PCR using following oligo combinations: full-length SND1 with oligos hSND1-F and hSND1-R; ΔC-SND1 with oligos hSND1_N-ter-F and hSND1_N-ter-R; ΔN-SND1 with oligos hSND1_C-ter-F and hSND1_C-ter-R; SN-domain SND1 with oligos hSND1_SN domain-F and hSND1_SN domain-R; TD domain SND1 with oligos hSND1_TD domain-F and hSND1_TD domain-R; NLS-SND1 with oligos hNLS-SND1 -F and hNLS-SND1 -R. Resulting DNA fragments were cloned into pCR8/GW/TOPO Gateway entry vector using TA Cloning Kit (Thermo Fisher, K250020). Lentiviral SND1 expression constructs were generated using Gateway LR cloning into pLenti6/V5-DEST vector. Plasmid with full-length human MTDH cDNA was gifted by Dr. Patrick Paddison (FHCC). Full-length MTDH was generated by PCR using oligos hMTDH-F and hMTDH-R. Resulting DNA fragments were cloned into pCR8/GW/TOPO Gateway entry vector using TA Cloning Kit (Thermo Fisher, K250020). Lentiviral MTDH expression constructs were generated using Gateway LR cloning into pLenti6/V5-DEST vector. All plasmids generated using PCR were sequence verified.

pGIPZ control, pGIPZ-shSND1#1 (V3LHS_300794), shSND1#2 (V3LHS_212892), shMTDH#1 (V3LHS_400003) and shMTDH#2 (V2LHS_118616) were purchased from Shared Resources Genomics

Center (FHCC). Human SND1 sgRNA target sequences were designed using the Broad Institute GPP sgRNA designer tool (portals.broadinstitute.org/gpp/public/analysis-tools/sgrna-design). sgRNA target sequences were subcloned into the pZHB-Z:U6_EFS-mCherry-Puro sgRNA vector using ESP3I sites and following annealed oligos: sgRNA#1 using oligos sgRNA#1(+) and sgRNA#1(-); sgRNA#2 using oligos sgRNA#2(+) and sgRNA#2(-). pZHB-Z:U6_EFS-mCherry-Puro sgRNA vector and non-targeting sgRNA control plasmid were a gift from Dr. Patrick Paddison (FHCC).

All siRNA oligos were purchased from Qiagen. siRNA oligos targeting human ERG: #1 custom with target sequence 5′-AACGACA TCCTTCTCTCACAT-3′ and #2 custom with target sequence 5′-CTCC ACGGTTAATGCATGCTA-3′, and #3 custom with target sequence 5′-GA TGATGTTGATAAAGCCTTA-3′. siRNA oligos targeting human SND1: #1 custom with target sequence 5′-CAGGCTGAACCTGTGGCGCTA-3′ and #2 custom with target sequence 5′-CAGCGTAGTTCGGGATATCCA-3′. siRNA oligos targeting human MTDH: #3 with target sequence 5′-TCC AGCCGAAGTACTCGTCAA-3′ and #4 with target sequence 5′-TGGGATGTTAGCCGTAATCAA-3′. AllStars Negative Control siRNA (Qiagen, Cat# 1027281) was used as control.

## Affinity purification and mass spectrometry

VCaP human prostate cancer cell lines stably expressing N- or C-terminal VA tagged ERG or GFP control proteins were generated using pLD-puro-CnVA-ERG, pLD-puro-CcVA-ERG and pLD-puro-CnVA-GFP lentiviral infection and puromycin selection, as described[62]. Expression of all proteins was Western-blot verified. Affinity purification using anti-Flag M2 antibody and purification was performed as previously described[63]. Briefly, briefly, protein lysates were generated by lysing two 15-cm plates of cells in TBS buffer (30 mM Tris-HCl at pH 7.5, 150 mM NaCl) containing 0.5% Nonidet P40 and protease and phosphatase inhibitors (Complete, Roche). Lysates were incubated with anti-Flag M2 antibody bound to Protein G Dynabeads for 2 h at 4 °C. Beads were washed three times, resuspended in 50 mM NH4HCO3, and incubated with sequencing-grade trypsin (Promega) overnight at 37 °C. Formic acid (2% final concentration) was added to terminate digestion, and the samples were desalted using C-18 cartridges (10–200 μL of NuTip; Glygen Corp.). Digested peptides were resolved on a microcolumn (120 mm × 75 μm) packed with 100 mm of 3-μm Luna C18 stationary phase (Phenomenex) using an organic gradient of 98% buffer A (5% acetonitrile, 0.1% formic acid) to 90% buffer B (95% acetonitrile 0.1% formic acid) over 45 min at a flow rate of 300 nL/min. Eluting peptides were electrosprayed directly into an Orbitrap Velos mass spectrometer (ThermoFisher Scientific). Spectra were analyzed using MaxQuant[64].

## RNA-Seq analyses

RNA concentration, purity, and integrity was assessed by NanoDrop (Thermo Fisher) and Agilent TapeStation. RNA-seq libraries were constructed from 1 μg total RNA using the Illumina TruSeq Stranded mRNA LT Sample Prep Kit according to the manufacturer's protocol. Barcoded libraries were pooled and sequenced on a NovaSeq S1 100 flowcell generating 50 bp paired-end reads. Sequencing reads were mapped to the hg38 human or mm10 mouse genomes using STAR.v2.7.3a1. Gene level abundance was quantitated using GenomicAlignments[65] and analyzed using limma[66], filtered for a minimum expression level using the filterByExpr function with default parameters prior to testing, and using the Benjamin-Hochberg false discovery rate (FDR) adjustment. Genome-wide gene expression results were ranked by their limma t-statistics and used to conduct Gene Set Enrichment Analysis (GSEA) to determine patterns of pathway activity utilizing the curated pathways from within the MSigDBv7.4[67].

## Tissue dissection, histology, and immunohistochemistry

For paraffin sections, the entire prostates were dissected, fixed in 4% formaldehyde in PBS overnight, processed and embedded in paraffin.

Sections (5 μm thick) were stained and imaged using a Nikon TE 200 microscope. For cryosections, tissues were frozen in OCT and sectioned (7 μm thick) using a Leica cryostat. For histology, sections were stained with hematoxilin & eosin. For immunohistochemistry, sections were deparaffinized, rehydrated, and antigenic sites were unmasked using either Tris-EDTA (10 mM Tris-HCl, pH 9.0; 1 mM EDTA; 0.05% Tween-20) or citric acid-based unmasking solution (Vector Laboratories) in Pascal pressure chamber (Dako). The sections were immunostained using EnVision and ARK kits (DAKO, K400311-2 and K395411-8) according to manufacturer protocols.

## Statistics and reproducibility

Data are presented as means +/− SD and graphs are generated using GraphPad Prism 10. Statistical significance was determined by unpaired, two-tailed Student's, ANOVA or Chi-square with Yates correction tests. Differences at $p = 0.05$ and lower were considered statistically significant. Number of times each experiment was conducted independently and sample size for each experiment are indicated in the legends. Sample sizes were chosen to ensure adequate statistical power based on the size of effects observed and reaching statistical significance based on the available samples. In RNA-seq analyses, the differences with $FDR < 0.05$ were considered statistically significant. The investigators were not blinded to allocation during experiments and outcome assessment.

## Reporting summary

Further information on research design is available in the Nature Portfolio Reporting Summary linked to this article.

## Data availability

RNA-Seq data generated in this study are available in the Gene Expression Omnibus repository (GEO) under accession number GSE212840. The mass spectrometry proteomics data are available in the PRIDE with the dataset identifier PXD036882. All remaining data can be found in the Article, Supplementary and Source data files. Source data are provided with this paper.

## Code availability

The code related to the Mass Spectrometry data analysis is publicly available at https://github.com/cnsb-boston/Omics_Notebook. The codes related to the RNA-Seq data analysis are publicly available at https://github.com/alexdobin/STAR, https://bioconductor.org/packages/release/bioc/html/GenomicAlignments.html, https://bioconductor.org/packages/release/bioc/html/limma.html, http://software.broadinstitute.org/gsea/index.jsp, and http://software.broadinstitute.org/gsea/msigdb. No code was generated specifically for this study.

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

## Acknowledgements
We thank Dr. J. K. Lee for help with establishing matrigel drop cultures, Dr. A.M. Chinnaiyan for sharing ERG constructs, Dr. H. Wu for sharing *Pten^{flox/flox}* mice, Dr. P. Paddison for gift of ORF and CRISPR/Cas9 plasmids, Dr. S. Beronja for gift of shRNA plasmids, and the Genomics Core Facility of FHCC for next generation sequencing. This research was supported in part by the Department of Defense Prostate Cancer Research grants W81XWH-20-1-0082, W81XWH-20-1-0111, NCI grants R01CA176844, R01CA234751, P01 CA163227, a developmental research project funded by 5P50CA097186, FHCC discretionary funds and NIH/NCI Cancer Center Support Grant P30 CA015704.

## Author contributions
Conceptualization: S-Y.L. and V.V.; methodology: S-Y.L., I.C., P.S.N., M.C.H., A.E., and V.V.; investigation: S-Y.L., D.R., S.B.F., O.K., J.K., L.T.P., I.C., D.L., and M.C.H.; writing—original draft: S-Y.L. and V.V.; writing—review & editing: all authors; funding acquisition: S-Y.L., P.S.N., A.E., and V.V.; supervision: A.E. and V.V.

## Competing interests
The authors declare no competing interests.
