## [Peer Review File · Nature Communications]

Reviewer #2:

Remarks to the Author:

The novelty of this study is that for the first time the authors showed that interaction of ERG, an ETS-domain transcriptional factor, and SND1 (via Tudor domain of SND1) plays a role in prostate cancer. In addition to that they have used new animal model PB-Cre4/PTENfl/fl/ERG/Snd1fl/fl to document that knocking out SND1 impedes ERG-induced prostate carcinogenesis. Although these findings are novel, there is a lack of mechanistic studies, and there are premature conclusions which significantly dampen enthusiasm for the paper.

Major concerns:

1. For co-IP analyses, C-terminal-tagged constructs were used for SND1 and MTDH, while N-terminal-tagged construct was used for ERG. N-terminal tags might interfere with subcellular localization of proteins. It would be important to perform co-IP and check subcellular localization of the interaction using a C-terminal-tagged ERG-HA construct.
2. SND1 and MTDH are exclusively in the cytoplasm while ERG is exclusively in the nucleus. Interaction between SND1 and ERG is not evident in Extended Data Fig. 1 (Merged image of blue/green/red creates white, no white dots are seen in the nucleus). Without demonstratable interaction among endogenous proteins the significance of the findings comes into question. Very little SND1 is present in the nucleus raising the concern whether that little amount of SND1 in the nucleus is able to modulate nuclear ERG function which is distributed evenly throughout the nucleus at a high level.
3. MTDH is an important protein. The authors' inference is that it is SND1 which interacts with ERG and MTDH is pulled down just because it interacts with SND1. It might be that SND1 directly interacts with ERG, but what is MTDH doing in this complex? If MTDH is bound to SND1 in SND1/ERG complex, how does MTDH modulate the function of this complex? Functional studies need to be done to address this issue. Extended Data Fig. 1 shows interaction of ERG with SND1 or MTDH separately. A tri-color IF needs to be done to check the interaction of these three proteins at subcellular level.
4. SND1 modulation (knocking down/knocking out/overexpression) did not have an effect on basal 3D and 2D colony formation in RWPE-1 cells. Considering the known role of SND1 in regulating proliferation and anchorage-independent growth, this finding is difficult to understand (especially when SND1 knockdown or overexpression showed an effect on basal proliferation of VCaP and LuCaP cells). Along this line, it might be worth comparing gene expression of RWPE-1 and VCaP cells upon SND1 knockdown. It is necessary to check the effect of si-MTDH on the basal and ERG-overexpressed 2D and 3D colonies of RWPE-1 cells. Additionally, invasion and proliferation of ERG-overexpressing RWPE-1 cells in the presence and absence of SND1 and MTDH should be checked.
5. From Fig. 5, the authors concluded that ERG promotes nuclear localization of SND1/MTDH. This conclusion is derived from overexpression studies. VCaP cells which have abundant nuclear ERG have little nuclear SND1/MTDH. So, the conclusion is not reflected in endogenous proteins. If SND1 with an NLS increases colony size of RWPE-1-SND1-1-/- cells, why is there no difference in colony size between RWPE-1 and RWPE-1-SND1-1-/- cells? What is the function of SND1 and MTDH in the nucleus? Are their functions dependent on ERG? Mechanistic studies are required to interrogate how ERG and SND1 regulate G2-M checkpoint genes (Fig. 4).
6. All experiments are done using ERG overexpression upon SND1 knockdown/knockout, documenting that ERG function requires SND1. Reverse experiments (i.e., using ERG-/- cells with SND1 overexpression) should have been done to check whether SND1 function requires ERG.
7. Fig. 7 is missing PB-Cre4/PTENfl/fl as a control to show that PB-Cre4/PTENfl/fl/ERG has more tumors than PB-Cre4/PTENfl/fl and this increase is decreased in PB-Cre4/PTENfl/fl/ERG/SND1fl/fl.
8. Fig. 4 and 8 identified genes that are modulated. But whether these genes play any functional role were not interrogated.

Minor concerns:

1. There are 89 and 104 unique proteins for Flag-ERG and ERG-Flag, respectively (Fig. 1b). This finding requires explanation. Additionally, there are 283 proteins common to Flag-GFP, Flag-ERG and ERG-Flag, raising concern about the stringency of the screen.
2. Fig2b: ERG level is increased in IP-SND1 sample with siMTDH. This is an important observation that needs better explanation.

3. In Western blots, indicating the molecular weight of the proteins would help better understanding.
4. MTDH/SND1 modulates immune factors. Checking immunological parameters in tumor-bearing mouse models would have provided novel and interesting finding.

Reviewer #3:

Remarks to the Author:

This is a well written and well constructed manuscript. Overall, I am strongly in support of publication at Nat. Com. I have a few suggestions for improvements that I hope do not present a significant burden, but in my view will meet a higher bar for quality and reproducibility, especially with respect to complicated genetically engineered mouse models.

The table of mice shows that some of the Snd1 f/f mice do develop invasive carcinomas, but these are not shown. It would be useful to the analysis to present (and compare phenotypes) of the invasive tumors here compared to control. Similarly, comparing in situ to in situ could be informative. One way to achieve this might be to add supplemental figures from this histology. If the journal will support WSI, that would be even better.

The trivial explanation (that should be considered and excluded) for reduced tumorigenesis in the Snd1 f/f is the strain background contribution. The statement in methods that all mice are 129 x B6 is somewhat dissatisfying. This has been the (low) standard in the literature, however, and I recognize that. Nevertheless, I would recommend a medium density snp panel (or similar) to assess the contributions per animal. In my view this should be standard for any mixed background strains autochthonous models.

I am interested in the mouse organoid phenotypes. The organoids (at least those shown in the images) appear to be plated at too high density. At this density it is impossible to know if there are effects related to "contact inhibition" for example. Interestingly, the organoid size seems to be the product of very well organized hollow spheroids. It would be great to have confocal or pelletized FFPE section morphology from these (and possible IHC for cell type contributions, proliferate rate, etc). Looking at the images available, there does seem to be some increase in size in the non- f/f ERG+ organoids... although clearly this observation does not reach statistical significance. Nevertheless, the statement heading "Mouse Snd1 is necessary for the ERG-mediated increase in prostate organoid size." should be modified. Necessity is not shown without a longer time course. i.e. Do the f/f ERG+ organoids continue to enlarge albeit slower than the wt/wt ERG+? Or do they never expand?

Finally, I must insist that the statements at the bottom of page 7: "Snd1 is necessary for prostate cancer growth in vivo." and "We concluded that while Snd1 is not necessary for normal prostate gland homeostasis, it is required for autochthonous prostate cancer development and growth." ... be revised. The data do not show "necessary" nor "required" but do show a difference in penetrance and/or rate of onset/initiation and/or rate of growth/progression. I'd be very interested in the details about this, which are not provided in the current manuscript.

Finally, I do not like the title. The use of the full official gene name for SND1 may be (an antiquated) policy of the journal, but is distracting. Furthermore, the words "interacts" with and "plays a critical role" could be made substantially more specific.

Point by point reply to reviewer's questions and comments.

We sincerely thank all the reviewers for the time they took to analyze this work and for their thoughtful comments and suggestions. We performed many experiments and modified the text according to these recommendations. All the modifications to the original manuscript are underlined. We feel that these changes significantly strengthened the study, and we are very grateful for the reviewer's assistance.

Reviewer #1

We are delighted that the reviewer indicated that our experiments “have appropriate controls to back up the finding” and that our “findings are important for our understanding of ERG-mediated tumorigenesis and should be of interest to both prostate cancer researchers, but also those interested in non-canonical roles of the SND1/MTDH proteins that normally regulate miRNAs.” We are grateful that the reviewer felt that “the manuscript is close to publication ready. The comments below are minor.”

Specifically, we have addressed the reviewer's comments in the following ways:

Minor comments:

Comment: “Be sure to include language that makes it clear that the data can only show indirect interaction between ERG and SND1/MTDH, as there are no experiments that test a direct interaction.”

Reply:

We agree with this important comment. We carefully went through the text of the manuscript to make sure that we are not claiming or implying that we discovered that ERG directly binds to SMD1/MTDH. While they form a protein complex in prostate cancer cells, it is formally possible that ERG-SND1 interaction is indirect.

Comment: “It would be helpful to mention in the results section description of Figure 2A that this experiment is being done in VCaP.”

Reply:

We added this information to the results section (page 4).

Comment: “ The statement at the top of page 6 indicating that ERG is necessary for regulation of proliferation in VCaP cells is overstated. The data demonstrate a role for SND1 in proliferation, but the data for ERG in proliferation does not reach statistical significance.”

Reply:

We agree that the decrease in VCaP cell proliferation upon transient transfection with siERG oligoes is not dramatic, but it is statistically significant (Fig. 3i). We modified the text to indicate that while the contribution of ERG in regulating VCaP cell growth is significant, we are not claiming that ERG is required for VCaP cell proliferation. We thank the reviewer for this comment.

Reviewer #2

We are pleased that the reviewer indicated that our “findings are novel”. The reviewer was not entirely convinced by our data and proposed several additional experiments that should strengthen the conclusions and increase the impact of the study. We are thankful for this reviewer's comments which helped tremendously to improve the manuscript. Specifically, we have addressed the reviewer's comments in the following ways:

Major concerns:

Comment 1: “ For co-IP analyses, C-terminal-tagged constructs were used for SND1 and MTDH, while N-terminal-tagged construct was used for ERG. N-terminal tags might interfere with subcellular localization of proteins. It would be important to perform co-IP and check subcellular localization of the interaction using a C-terminal-tagged ERG-HA construct.”

Reply:

We thank the reviewer for this important comment. We performed additional cloning experiments and generated C-terminal-tagged ERG-HA construct. We found that the C-terminal-tagged ERG-HA performed very similar to N-terminally tagged HA-ERG and both strongly interacted with SND1/MTDH protein complex (Fig. 1d). We also performed co-IP experiments with endogenous ERG, SND1 and MTDH proteins and revealed similar interaction (Fig. 1e).

Comment 2: “SND1 and MTDH are exclusively in the cytoplasm while ERG is exclusively in the nucleus. Interaction between SND1 and ERG is not evident in Extended Data Fig. 1 (Merged image of blue/green/red creates white, no white dots are seen in the nucleus). Without demonstratable interaction among endogenous proteins the significance of the findings comes into question. Very little SND1 is present in the nucleus raising the concern whether that little amount of SND1 in the nucleus is able to modulate nuclear ERG function which is distributed evenly throughout the nucleus at a high level.”

Reply:

We believe that our data indicate that it is not SND1 that modulates ERG function in the nucleus, but rather ERG interaction brings a small amount of SND1 to the nucleus in ERG expressing cells. It is this nuclear SND1 (protein with well documented transcription activation function) that plays a critical role in our model (Fig. 5c-e). Our data clearly indicate that endogenous ERG and SND1/MTDH interact with each other (Fig. 1e-f) and ERG promotes nuclear localization of SND1 (Fig. 5a). We found that the majority of SND1/MTDH remains in the cytoplasm even in ERG expressing cells (Extended data Fig. 1). We do not see a conflict with this finding. There are many precedents where nuclear translocation of a small proportion of a largely cytoplasmic protein that has transcription activation function results in profound changes and cellular transformation. For example, this is definitely the case with beta-catenin. While the overwhelming majority of beta-catenin is always in the cytoplasm, a small increase in nuclear beta-catenin has tremendous impact on cellular physiology and often results in cancer.

Concerning the comment that the “*Merged image of blue/green/red creates white*”. We believe that this happens only when the intensity of all three colors is exactly the same. When one color is significantly more intense than the others, this color will dominate and there will be no white created in the combined image.

Comment 3: “MTDH is an important protein. The authors’ inference is that it is SND1 which interacts with ERG and MTDH is pulled down just because it interacts with SND1. It might be that SND1 directly interacts with ERG, but what is MTDH doing in this complex? If MTDH is bound to SND1 in SND1/ERG complex, how does MTDH modulate the function of this complex? Functional studies need to be done to address this issue. Extended Data Fig. 1 shows interaction of ERG with SND1 or MTDH separately. A tri-color IF needs to be done to check the interaction of these three proteins at subcellular level.”

Reply:

We purchased guinea pig anti-SND1 antibodies and performed the tri-color IF as suggested by the reviewer (Extended data Fig. 1, lower panels). As we expected, MTDH displays very similar staining pattern compared to the SND1 staining pattern.

We agree that MTDH is an important protein, but because we found that ERG primarily binds to SND1, we concentrated on physiological functions of SND1. In addition, based on genetic studies from the Dr. Yibin Kang laboratory the primary function of MTDH is to stabilize SND1 and it is SND1 that plays a critical functional role in MTDH/SND1-mediated transformation (Wan et al., Cancer Cell, 2014). We expected that the functional data for MTDH would be similar to SND1. Indeed, we now show that cells with knockdown of

endogenous MTDH have very similar phenotypes to cells with knockdown of endogenous SND1 (Extended data Fig. 2).

Comment 4: “SND1 modulation (knocking down/knocking out/overexpression) did not have an effect on basal 3D and 2D colony formation in RWPE-1 cells. Considering the known role of SND1 in regulating proliferation and anchorage-independent growth, this finding is difficult to understand (especially when SND1 knockdown or overexpression showed an effect on basal proliferation of VCaP and LuCaP cells). Along this line, it might be worth comparing gene expression of RWPE-1 and VCaP cells upon SND1 knockdown. It is necessary to check the effect of si-MTDH on the basal and ERG-overexpressed 2D and 3D colonies of RWPE-1 cells. Additionally, invasion and proliferation of ERG-overexpressing RWPE-1 cells in the presence and absence of SND1 and MTDH should be checked.”

Reply:

We believe that our data are showing that SND1/MTDH are largely dispensable in normal cells but play an important role in cancer cells. This is consistent with the literature. MTDH knockout mice are normal and display the difference only upon cancer initiation (Wan et al., Cancer Cell, 2014, Shen et al., Nature Cancer, 2022). Moreover, our data with the newly generated conditional mutant allele of *Snd1* show that *Snd1*^{-/-} prostate is completely normal; however, there is a tremendous impact in the genetic model of prostate cancer. (Figs. 5-6). RWPE-1 cells are normal epithelial cells, while VCaP and LuCaP are cancer cells. Normal and cancer cells have vastly different physiologies and transcriptional programs. It is not surprising that SND1/MTDH play a different role in normal versus transformed cells. In the original manuscript, we showed data quantitating proliferation in RWPE-1-Ctrl and RWPE-1-ERG cells with knockdown of SND1 (Fig. 3a-c), and now add the data with knockdown of MTDH (Extended data Fig. 2). The data are very similar.

Comment 5: “From Fig. 5, the authors concluded that ERG promotes nuclear localization of SND1/MTDH. This conclusion is derived from overexpression studies. VCaP cells which have abundant nuclear ERG have little nuclear SND1/MTDH. So, the conclusion is not reflected in endogenous proteins. If SND1 with an NLS increases colony size of RWPE-1-SND1-1^{-/-} cells, why is there no difference in colony size between RWPE-1 and RWPE-1-SND-1^{-/-} cells? What is the function of SND1 and MTDH in the nucleus? Are their functions dependent on ERG? Mechanistic studies are required to interrogate how ERG and SND1 regulate G2-M checkpoint genes (Fig. 4).”

Reply:

We believe that our data show that while ERG promotes nuclear localization of SND1/MTDH, only small part of the entire cellular pool of SND1/MTDH is responding and the majority of the protein complex remains in the cytoplasm. Exactly why this happens and what protein complexes retain SND1/MTDH in the cytoplasm is presently unknown. However, this is not unusual. For example, overexpression of LEF/TCF proteins translocates only a small proportion of beta-catenin to the nucleus, however, this small amount of nuclear beta-catenin has a tremendous physiological impact. We know that interaction with cadherins and alpha-catenin retains the bulk of beta-catenin in the cytoplasm. Presently, we do not know what regulates the cellular localization of SND1/MTDH, but this will be very likely discovered in the future. However, this is well beyond the scope of this particular investigation.

Indeed, NLS-SND1 increases colony size of RWPE-1-SND1-1^{-/-} cells, but there no difference in colony size between RWPE-1 and RWPE-1-SND-1^{-/-} cells. This is completely consistent with our model. SND1 has no obvious function in normal cells (no difference in colony size between RWPE-1 and RWPE-1-SND-1^{-/-} cells), but even a small amount of forced nuclear NLS-SND1 has a tremendous impact, and this is not dependent on ERG, because these cells do not express ERG. It is possible that forced nuclear NLS-SND1 becomes an oncogene. Exactly what function it has in the nucleus is not completely established. One possibility is very obvious: SND1 is a well-known transcriptional co-activator and this is how it may regulate the transcription of G2-M checkpoint genes. Future studies will help to investigate this in more detail.

Comment 6: “All experiments are done using ERG overexpression upon SND1 knockdown/knockout, documenting that ERG function requires SND1. Reverse experiments (i.e., using ERG^{-/-} cells with SND1 overexpression) should have been done to check whether SND1 function requires ERG.”

Reply:

We performed the requested experiments. Fig. 3l. Overexpression of SND1 increases proliferation of VCaP ERG-positive prostate cancer cells, and the knockdown of ERG significantly downregulates the proliferation of these cells.

Comment 7: “ Fig. 7 is missing PB-Cre4/PTEN^{fl/fl} as a control to show that PB-Cre4/PTEN^{fl/fl}/ERG has more tumors than PB-Cre4/PTEN^{fl/fl} and this increase is decreased in PB-Cre4/PTEN^{fl/fl}/ERG/SND1^{fl/fl}.”

Reply:

We agree that the generation and analysis of an additional mouse model would extend the study. However, generation of conditional allele of *Snd1* in mice and its characterization represented a very substantial effort and a long time-frame (years) to generate the results. Upon characterization of mice with conditional knockout of *Snd1* in the prostate gland and finding no obvious phenotype, we decided to concentrate on the most relevant genetic model of prostate cancer. We chose to use PB-Cre/PTEN^{fl/fl}/ERG mice, because overexpression of ERG and loss of PTEN occurs in a very large proportion of human prostate cancer. It was important to know whether targeting SND1 represents a viable avenue for the treatment of patients with these tumors. In this mouse model we found a tremendous impact resulting from the ablation of SND1 on tumor growth and we would like to report this finding. We are very careful not to claim that SND1 is required only in ERG⁺ prostate cancer in mice, because presently we do not have these data. Future experiments with additional genetic models of prostate cancer will answer this question, but this is beyond the scope of this initial study on the conditional allele of SND1.

Comment 8: Fig. 4 and 8 identified genes that are modulated. But whether these genes play any functional role were not interrogated.

Reply:

ERG and SND1 regulate the expression of many genes. We hope that the reviewer agrees that the analysis of the role of all these genes is beyond the scopes of this study.

Minor concerns:

Comment 1: “There are 89 and 104 unique proteins for Flag-ERG and ERG-Flag, respectively (Fig. 1b). This finding requires explanation. Additionally, there are 283 proteins common to Flag-GFP, Flag-ERG and ERG-Flag, raising concern about the stringency of the screen.”

Reply:

These numbers are consistent with the prior experience of our proteomics expert Dr. Andrew Emili (Kwan et al., Genes&Dev, 2016; Babu et al., Nat Biotech, 2018; Marcon et al., Cell Rep, 2014; Babu et al., Nature, 2012). In addition to specific interactors, protein pull-down mass spectrometry experiments generate a lot of non-specifically bound background proteins. Proteins that bind to protein tag, to resin, or proteins that are very abundant and easy to identify on mass spectrometer. Utilization of GFP as a negative control and both N-terminally and C-terminally tagged proteins helps to weed out these proteins and reveal specific interactors. Even after all this, it is still possible to have a false positive. Therefore, the initial screen is always followed by co-IP experiments, to confirm the interaction. This is exactly what we did in our Figs 1-2.

Comment 2. *“Fig2b: ERG level is increased in IP-SND1 sample with siMTDH. This is an important observation that needs better explanation.”*

Reply:

The levels of ERG in IP-SND1 samples in Fig. 2b are consistent with the levels of pulled down SND1 shown in the same figure.

Comment 3: *“In Western blots, indicating the molecular weight of the proteins would help better understanding.”*

Reply:

We added the positions of molecular weight markers, where space permitted.

Comment 4: *“MTDH/SND1 modulates immune factors. Checking immunological parameters in tumor-bearing mouse models would have provided novel and interesting finding.”*

Reply:

We agree that this is a potentially interesting direction to explore. However, this is well beyond the scope of the current manuscript.

Reviewer #3

We are pleased that the reviewer found that we have “a well written and well constructed manuscript” and indicated a “strong support of publication at Nat. Com.” We found that the comments and suggestions of the reviewer were very helpful. The resulting changes tremendously strengthened and improved the impact of our work. Specifically, we have addressed the reviewer’s comments in the following ways:

Comment #1: *“The table of mice shows that some of the Snd1 f/f mice do develop invasive carcinomas, but these are not shown. It would be useful to the analysis to present (and compare phenotypes) of the invasive tumors here compared to control. Similarly, comparing in situ to in situ could be informative. One way to achieve this might be to add supplemental figures from this histology. If the journal will support WSI, that would be even better.”*

Reply:

We are thankful for this suggestion. We added two supplementary figures (Figs. 5-6) to show the invasive carcinomas and in situ lesions, as well as additional examples of whole-mount prostate histologies from multiple control *PB-Cre/Pten^{f/f}/ERG* and *Snd1*-mutant *PB-Cre/Pten^{f/f}/ERG/Snd1^{f/f}* mice.

Comment #2: *“The trivial explanation (that should be considered and excluded) for reduced tumorigenesis in the Snd1 f/f is the strain background contribution. The statement in methods that all mice are 129 x B6 is somewhat dissatisfying. This has been the (low) standard in the literature, however, and I recognize that. Nevertheless, I would recommend a medium density snp panel (or similar) to assess the contributions per animal. In my view this should be standard for any mixed background strains autochthonous models.”*

Reply:

We agree that while this is not yet practiced in the current literature, this will and should become the standard in the future. Unfortunately, we did not think of that at the time when we collected the experimental animals, and we presently do not have their DNA samples for this analysis. Since the control *PB-Cre/Pten^{f/f}/ERG* and *Snd1*-mutant *PB-Cre/Pten^{f/f}/ERG/Snd1^{f/f}* mice are generated as littermates coming from the same breeding set up,

we feel that it is extremely unlikely that the mice with final genotypes have a consistently different genetic background.

Comment #3: "I am interested in the mouse organoid phenotypes. The organoids (at least those shown in the images) appear to be plated at too high density. At this density it is impossible to know if there are effects related to "contact inhibition" for example. Interestingly, the organoid size seems to be the product of very well organized hollow spheroids. It would be great to have confocal or pelletized FFPE section morphology from these (and possible IHC for cell type contributions, proliferate, etc).

Reply:

For mouse organoid experiments we are using a newly developed Matrigel drop culture system. We describe it in methods section. In this approach, Matrigel is mixed with single cell suspension and deposited as 15µl drops in the middle of each well of 48-well plate. This helps tremendously minimize the amount Matrigel used for each sample/time point and to increase the overall number of replicates for each sample/time point. We tried different cell densities in this method and now using the density that seems to work best in our hands. The Matrigel drop culture was developed by Dr. John Lee at the Fred Hutch and he will publish in the future the data on cell type contribution, proliferation rates etc. for this method. In our study, we used exact same conditions for both control and *Snd1*-mutant cells. We hope that the reviewer agrees that while the data on cell type contribution, proliferation rates etc. for drop cell culture may be very interesting, this is beyond the scopes of our current manuscript.

Comment #4: "Looking at the images available, there does seem to be some increase size in the non- f/f ERG+ organoids... although clearly this observation does not reach statistical significance. Nevertheless, the statement heading "Mouse Snd1 is necessary for the ERG-mediated increase in prostate organoid size." should be modified. Necessity is not shown without a longer time course. i.e. Do the f/f ERG+ organoids continue to enlarge albeit slower than the wt/wt ERG+? Or do they never expand?"

Reply:

The difference between wild-type and *ERG*+ organoids is highly statistically significant (Fig. 6m). This difference in size is completely erased when *Snd1* is deleted (Fig. 6m). We believe that these data indicate that *Snd1* was required to observe the difference in size between the wild-type and *ERG*+ organoids in the framework of the experiment. However, we agree with the reviewer that our text describing the results was too assertive and our conclusions too general. Therefore, we changed the text to describe the exact data that we obtained without any generalization and using such terms as "required" or "necessary".

Comment #4: "Finally, I must insist that the statements at the bottom of page 7: "Snd1 is necessary for prostate cancer growth in vivo." and "We concluded that while Snd1 is not necessary for normal prostate gland homeostasis, it is required for autochthonous prostate cancer development and growth." ... be revised. The data do not show "necessary" nor "required" but do show a difference in penetrance and/or rate of onset/initiation and/or rate of growth/progression. I'd be very interested in the details about this, which are not provided in the current manuscript.

Finally, I do not like the title. The use of the full official gene name for SND1 may be (an antiquated) policy of the journal, but is distracting. Furthermore, the words "interacts" with and "plays a critical role" could be made substantially more specific."

Reply:

Similar to the comment/reply #4, we modified the text to remove the terms "necessary", "required", "interacts" and "plays a critical role" and replaced them with more simple and straightforward description of the results obtained in this study. All the changes to the title and text of the manuscript are underlined.

Reviewers' Comments:

Reviewer #1:

Remarks to the Author:

The authors have satisfied all of my concerns.

The authors also satisfied most of the concerns of Reviewer #3. The authors added additional figures to fully satisfy comment #1. The authors were not able to fully address comment #2, however the reviewer did concede that the current controls met the standard of the field. The authors were able to address comments 3,4, and 5 through modifications of the text. These modifications were responsive and fully satisfy these concerns.

Reviewer #2:

Remarks to the Author:

The authors have successfully addressed the previous concerns.

Point by point response to the reviewer's comments.

We sincerely thank all the reviewers for their comments.

Reviewer #1 (Remarks to the Author):

The authors have satisfied all of my concerns.

The authors also satisfied most of the concerns of Reviewer #3. The authors added additional figures to fully satisfy comment #1. The authors were not able to fully address comment #2, however the reviewer did concede that the current controls met the standard of the field. The authors were able to address comments 3,4, and 5 through modifications of the text. These modifications were responsive and fully satisfy these concerns.

Reply: We are happy that we were able to satisfy all the concerns. We thank the reviewer for time and effort.

Reviewer #2 (Remarks to the Author):

The authors have successfully addressed the previous concerns.

Reply: We are pleased that we satisfied all the concerns. We thank the reviewer for time and effort.